# Structural dynamics of *E. coli* single-stranded DNA binding protein reveal DNA wrapping and unwrapping pathways

Sukrit Suksombat[1], Rustem Khafizov[1], Alexander G Kozlov[2], Timothy M Lohman[2]*, Yann R Chemla[1]*

[1]Department of Physics, Center for the Physics of Living Cells, Center for Biophysics and Computational Biology, University of Illinois at Urbana-Champaign, Urbana, United States; [2]Department of Biochemistry and Molecular Biophysics, Washington University School of Medicine, St. Louis, United States

**Abstract** *Escherichia coli* single-stranded (ss)DNA binding (SSB) protein mediates genome maintenance processes by regulating access to ssDNA. This homotetrameric protein wraps ssDNA in multiple distinct binding modes that may be used selectively in different DNA processes, and whose detailed wrapping topologies remain speculative. Here, we used single-molecule force and fluorescence spectroscopy to investigate *E. coli* SSB binding to ssDNA. Stretching a single ssDNA-SSB complex reveals discrete states that correlate with known binding modes, the likely ssDNA conformations and diffusion dynamics in each, and the kinetic pathways by which the protein wraps ssDNA and is dissociated. The data allow us to construct an energy landscape for the ssDNA-SSB complex, revealing that unwrapping energy costs increase the more ssDNA is unraveled. Our findings provide insights into the mechanism by which proteins gain access to ssDNA bound by SSB, as demonstrated by experiments in which SSB is displaced by the *E. coli* recombinase RecA.

*For correspondence: lohman@biochem.wustl.edu (TML); ychemla@illinois.edu (YRC)

**Competing interests:** The authors declare that no competing interests exist.

**Reviewing editor**: John Kuriyan, Howard Hughes Medical Institute, University of California, Berkeley, United States

## Introduction

*Escherichia coli* single-stranded DNA binding protein (*Eco*SSB) is an essential protein involved in most aspects of genome maintenance (*Meyer and Laine, 1990*; *Lohman and Ferrari, 1994*; *Shereda et al., 2008*). It binds with high affinity and little sequence specificity (*Lohman and Overman, 1985*; *Lohman and Ferrari, 1994*) to single stranded (ss)DNA intermediates formed during DNA replication, recombination, and repair, protecting them from both nucleolytic and chemical damage. SSB also interacts directly with more than a dozen proteins involved in genome maintenance, regulating their access to ssDNA and bringing them to their sites of action (*Shereda et al., 2008*).

*Eco*SSB is one of the most extensively studied ssDNA binding proteins. It consists of four identical subunits (~19 kDa each) that form a functional tetramer (*Raghunathan et al., 1997*, *2000*) (*Figure 1A*) that is stable over a wide range of solution conditions and at sub-nanomolar protein concentrations (*Lohman and Overman, 1985*; *Bujalowski and Lohman, 1991b*). Each monomer contains an oligonucleotide/oligosaccharide binding (OB) fold that contains the ssDNA binding site (*Raghunathan et al., 2000*). Thermodynamic studies have shown that *Eco*SSB tetramers bind and wrap ssDNA in a variety of binding modes that differ primarily in the number of OB folds that interact with the tetramer (*Lohman and Ferrari, 1994*). Three different binding modes have been identified on poly(dT) at 25°C, termed $(SSB)_{65}$, $(SSB)_{56}$ and $(SSB)_{35}$, which occlude 65, 56, and 35 nucleotides (nt) per tetramer, respectively, with a fourth mode observed at 37°C that occludes 40 nt (*Bujalowski and Lohman, 1986*). These modes can reversibly interconvert, with the transitions influenced primarily by salt concentration and type as well as protein binding density on the DNA (*Bujalowski and Lohman, 1986*).

**eLife digest** The DNA double helix consists of two strands coiled around each other. However, there are many instances when DNA must be separated into its individual strands—for example, when the DNA sequence needs to be copied. These single-stranded structures are highly prone to damage. For protection, the single-stranded DNA can wrap around single-stranded DNA binding (SSB) proteins, which also control how other maintenance proteins interact with the DNA.

SSB proteins from the bacteria species *Escherichia coli* wrap single-stranded DNA into a variety of topologies known as binding modes. By using a technique that uses a laser to exert forces on an individual DNA molecule, Suksombat et al. unraveled DNA from a single SSB protein. This revealed that the unraveling occurs in a series of steps that correspond well to the known binding modes. These steps also provide the energies required to unravel the single-stranded DNA. Further experiments showed that SSBs can slide along DNA without having to change their binding mode.

The unraveling and sliding mechanisms are likely to be used by other proteins to gain access to DNA coated with SSBs. The next step is to understand how SSBs interact with these other proteins, and how their various wrapping configurations affect this interaction.

The $(SSB)_{35}$ mode also binds ssDNA with high cooperativity, forming protein clusters (*Sigal et al., 1972*; *Ruyechan and Wetmur, 1975*; *Lohman et al., 1986*; *Kozlov et al., 2015*) that may be important during DNA replication (*Lohman et al., 1988*). It has been suggested that SSB utilizes all of these binding modes during its different roles in genome maintenance (*Lohman et al., 1988*) and that transitions between modes may control access of other proteins to the ssDNA (*Wessel et al., 2013*; *Bhattacharyya et al., 2014*).

Crystallographic studies of a C-terminal truncation of the SSB tetramer (SSBc) with two molecules of $(dC)_{35}$ bound suggest a model for the $(SSB)_{65}$ mode in which 65 nt of ssDNA wrap around an SSB tetramer in a topology resembling the seams on a baseball (*Raghunathan et al., 2000*) (*Figure 1A*). Based on this structure, a model for the $(SSB)_{35}$ mode has also been proposed (*Raghunathan et al., 2000*). Less is known about the wrapping configurations of the other binding modes, especially the $(SSB)_{56}$ mode that has only been detected on long poly(dT) ssDNA (*Bujalowski and Lohman, 1986*). However, various techniques such as electron microscopy (*Chrysogelos and Griffith, 1982*; *Griffith et al., 1984*), SSB fluorescence quenching (*Lohman and Overman, 1985*; *Bujalowski and Lohman, 1986*, *1989a*, *1989b*; *Lohman et al., 1986*) and sedimentation (*Bujalowski et al., 1988*) have provided some basic constraints.

Recent single-molecule studies have provided new insights on SSB-ssDNA complex dynamics. Single-molecule FRET (smFRET) measurements characterized transitions between binding modes (*Roy et al., 2007*) and established that *Eco*SSB tetramers can diffuse along ssDNA (*Roy et al., 2009*) by a reptation mechanism (*Zhou et al., 2011*). Force spectroscopy approaches have also proven useful in studying single-stranded DNA binding protein interactions with DNA (*Pant et al., 2005*; *Shokri et al., 2006*; *Hatch et al., 2007*, *2008*). Force not only adds another variable to perturb protein-DNA interactions but also provides a well-defined reaction coordinate to quantify the energy landscape governing those interactions. Using a combination of optical traps and smFRET, *Zhou et al. (2011)* showed that force gradually unravels ssDNA from *Eco*SSB and proposed that the energy landscape for SSB-ssDNA interactions is smooth, with few barriers to unwrapping.

Here, we present direct observations of a single *Eco*SSB tetramer interacting with ssDNA using force spectroscopy combined with single-molecule fluorescence microscopy. Applying mechanical force to destabilize the SSB-ssDNA complex and facilitate transitions between binding modes, we show that the ssDNA exhibits discrete wrapping states consistent with the known $(SSB)_{65}$, $(SSB)_{56}$ and $(SSB)_{35}$ binding modes. Our results are compatible with putative models of the $(SSB)_{35}$ structure (*Raghunathan et al., 2000*) and reveal a likely wrapping configuration for the $(SSB)_{56}$ mode. SSB-$(dT)_{70}$ complexes exhibit reversible force-induced transitions between modes without dissociation and SSB can diffuse along ssDNA in the different binding modes, indicating a highly dynamic complex. The data also reveal details of the energy landscape for SSB-ssDNA interactions. In contrast to previous suggestions (*Zhou et al., 2011*), the landscape contains multiple barriers between discrete wrapping conformations, suggesting a distinct wrapping pathway for *Eco*SSB. Moreover, the energy density is

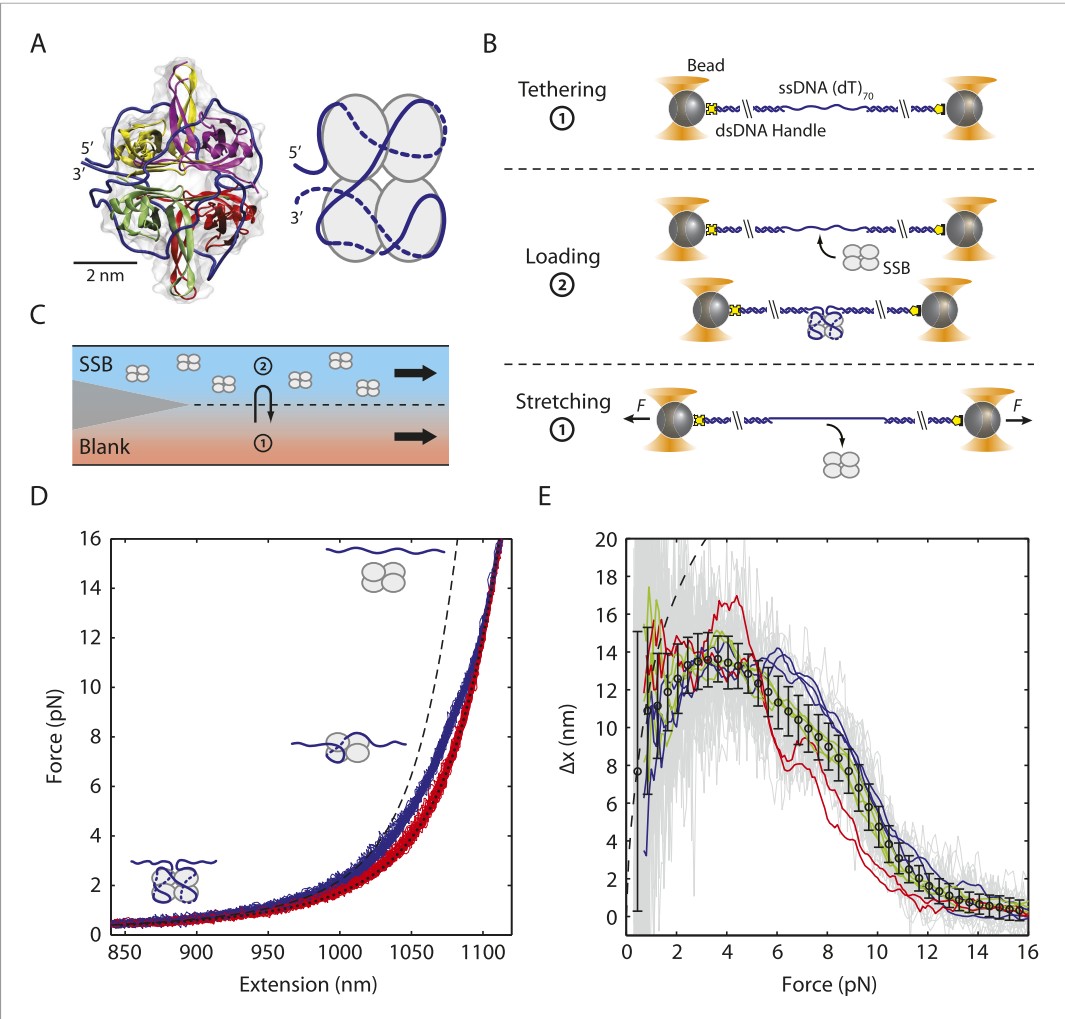

**Figure 1**. Unwrapping of ssDNA from *Escherichia coli* SSB under mechanical tension. (**A**) Crystal structure (Protein Data Bank ID number 1EYG) and schematic representation of an *E. coli* SSB tetramer wrapped by 70 nt of ssDNA (blue) in the $(SSB)_{65}$ mode. From 5′ to 3′, ssDNA interacts with the yellow, purple, green and red subunits. (**B**) Schematic of SSB unwrapping experiment. A DNA construct consisting of two long double-stranded DNA (dsDNA) handles and a short $(dT)_{70}$ ssDNA site is tethered between two optically trapped beads in the absence of SSB (Position 1, panel **C**). When moved to the stream containing SSB (Position 2), a single SSB tetramer binds to the ssDNA site at low tension (~0.5 pN). The tethered DNA is moved back to the blank stream (Position 1) and a ramping force is applied. Stretching the nucleoprotein complex to >20 pN causes the SSB to dissociate. (**C**) Experimental flow chamber. Two separate streams containing experimental buffer only (red, Position 1) and buffer plus 0.5 nM SSB (blue, Position 2) form a laminar interface with minimal mixing. (**D**) Representative force-extension curves (FECs). Relaxing curves (red) are obtained after SSB dissociation, and are well fit to a polymer model of bare DNA (black dotted line, 'Materials and methods'). Stretching curves (purple) of the SSB-ssDNA complex deviate from a model assuming the protein adopts the $(SSB)_{65}$ wrapping mode at all forces (black dashed line). Cartoon illustration of SSB unwrapping shows the SSB behavior at particular forces. (**E**) Change in extension upon SSB wrapping vs applied force. The change in extension is determined from the extension difference between stretching and relaxing curves in (**D**). Individual traces (gray) are binned and averaged to yield a mean change in extension (black opened circle; error bars are S.D.). The data deviates from the model (dashed line, determined from the difference between the dashed and dotted lines in (**D**)) at forces >1 pN. Representative traces (red, green, and blue) display the differences between the individual and averaged traces.

The following figure supplements are available for figure 1:

**Figure supplement 1**. Dissociation of SSB upon DNA stretching.

*Figure 1. continued on next page*

*Figure 1. Continued*

**Figure supplement 2**. Single-stranded DNA polymer modeling.

**Figure supplement 3**. Dissociation force of SSB-ssDNA.

**Figure supplement 4**. Sample chamber.

**Figure supplement 5**. DNA construct.

**Figure supplement 6**. SSB binds to $dT_{70}$ in the fully wrapped $(SSB)_{65}$ mode at a 1:1 molar ratio in 100 mM Tris buffer.

unbalanced, such that the energy cost of unwrapping increases as ssDNA is unraveled from its ends. These findings along with studies of the competition between *E. coli* SSB and the RecA recombinase protein demonstrate how SSB bound in its different modes might regulate accessibility to ssDNA of other genome maintenance proteins.

## Results

### Force unravels ssDNA from a single SSB tetramer

We used dual trap optical tweezers to stretch a SSB-ssDNA complex mechanically. As shown in *Figure 1B*, two trapped functionalized micron-sized beads were tethered together by a DNA construct consisting of a 70-nt poly(dT) ssDNA segment flanked by two long double-stranded DNA (dsDNA) 'handles' ('Materials and methods'). The length of the ssDNA was chosen to accommodate one SSB tetramer in its $(SSB)_{65}$ binding mode. We also worked under salt conditions and protein concentrations known to favor the $(SSB)_{65}$ mode in the absence of mechanical tension (*Bujalowski and Lohman, 1986*; *Roy et al., 2007*) ('Materials and methods'). Force-extension curves (FECs) of this construct in the absence of protein (*Figure 1—figure supplement 1*, green) were in excellent agreement with theoretical models of DNA elasticity ('Materials and methods'; *Figure 1—figure supplement 1*, black dashed line). The total extension of the 'bare' DNA molecule, $x_{bare}$, is given by the sum of the extensions of the dsDNA handles and the ssDNA binding site at a tension *F*:

$$x_{bare}(F) = \xi_{ds}(F) \cdot N_{ds} + \xi_{ss}(F) \cdot N_{ss}, \tag{1}$$

where $\xi_{ds}(F)$ and $\xi_{ss}(F)$ are the extension of one dsDNA base pair and one ssDNA nucleotide given by the extensible worm-like chain (XWLC, *Bustamante et al., 1994*) and 'snake-like chain (SLC)' model (*Saleh et al., 2009*), respectively ('Materials and methods'; *Figure 1—figure supplement 2*). $N_{ds} = 3260$ bp is the total length of the dsDNA handles and $N_{ss} = 70$ nt is that of the ssDNA loading site.

To investigate a single SSB tetramer-ssDNA complex, protein in solution was added to the construct ('Materials and methods'; *Figure 1B,C*) for a short period of incubation, allowing one SSB to bind the 70-nt ssDNA. The molecule was then stretched in the absence of free proteins in solution (*Figure 1B,C*). FECs of stretching and relaxing many molecules are shown in *Figure 1D*. The stretching FECs (violet) of the SSB-DNA complex displayed a shorter extension compared to those without protein due to ssDNA compaction by the SSB. Upon stretching to a force >20 pN and relaxing the molecule, the FECs (*Figure 1*, red) matched those in the absence of protein (*Figure 1—figure supplement 1*, green), indicating that the SSB had dissociated during the stretching process. We confirmed that a single SSB was loaded onto the DNA and dissociated at high force through simultaneous fluorescence detection of dye labeled protein. Using an instrument combining optical traps with a single-molecule fluorescence confocal microscope (*Comstock et al., 2011*), we detected SSB site-specifically labeled with an average of one AlexaFluor555 fluorophore ($SSB_f$) as we obtained a FEC (*Figure 1—figure supplement 3*; 'Materials and methods'). The average dissociation force was $10.3 \pm 0.9$ pN, consistent with previous reports (*Zhou et al., 2011*). Integrating the area between protein-bound and bare FECs to the force at which the complex spends half its time bound and half unbound yielded a value for the SSB-ssDNA wrapping free energy of $22 \pm 2$ $k_BT$ ('Materials and methods') similar to a previously reported value (*Zhou et al., 2011*).

The difference in extension between stretching and relaxing FECs provides information on the SSB-ssDNA wrapping conformation as a function of force. For SSB-bound DNA, we first considered that SSB adopted the canonical $(SSB)_{65}$ structure (*Raghunathan et al., 2000*). We thus expected a FEC given by *Equation 1* with $N_{ss} = 70 - 65 = 5$ nt due to occlusion by the SSB. As shown in *Figure 1D*, the stretching FECs (violet) diverged significantly from this theoretical model (black dashed line). *Figure 1E* displays the extension difference, $\Delta x$, between the stretching and corresponding relaxing curves as a function of tension $F$, averaged over many molecules ($N = 36$; black points), and the corresponding theoretical model (black dashed line). The agreement between model and data at tensions <1 pN is consistent with 65 nt being wrapped around SSB at low forces. Beyond this force, however, $\Delta x$ is consistently below the prediction, indicating that the SSB wraps <65 nt of ssDNA, in agreement with earlier measurements (*Zhou et al., 2011*).

Interestingly, neither the data in *Figure 1E* nor in those previous studies (*Zhou et al., 2011*) provide evidence for discrete wrapping morphologies such as $(SSB)_{56}$ and $(SSB)_{35}$ as observed in ensemble studies. If different SSB modes are stable and interconvertible, discrete transitions in the extension would have been expected in the stretching-relaxing experiment. However, detecting intermediates would be possible only if the rate at which the force was ramped was slower than the transitions between intermediates. Moreover, averaging over multiple molecules here and in *Zhou et al. (2011)* likely conceals transitions between SSB-ssDNA wrapping intermediates. Example individual traces (*Figure 1E*, blue, red, and green curves) support this view by illustrating the variability among FECs and their divergence from the average behavior (black). Rips in some of these traces (for example, the red traces at 5 pN) suggest that SSB may undergo transitions between different wrapping states.

## SSB binds ssDNA in intermediate wrapping states under tension

To investigate the presence of intermediate wrapping states further, we measured binding of individual SSB tetramers to the ssDNA at constant tension by operating the optical trap in a force-clamp mode ([*Neuman and Block, 2004*], 'Materials and methods'). As shown in *Figure 2A*, a DNA construct was initially held in the optical tweezers at a desired constant tension (2–10 pN) and protein was added. After a short time, an SSB binds, and the DNA is compacted upon wrapping. At the end of each observation, protein was dissociated by increasing the tension to a force (~25 pN) at which SSB cannot remain stably bound. This cycle was repeated numerous times to monitor new protein binding to the same DNA construct.

*Figure 2B* shows the change in DNA end-to-end extension, $\Delta x$, upon binding of SSB as a function of force. Using bare DNA as a reference (set to 0 nm), negative extension changes correspond to ssDNA wrapping and positive changes to release of wrapped DNA. At low tensions (<3 pN), we observed that individual SSBs bind and compact ssDNA in a single step (*Figure 2B*). SSBs remained bound to the ssDNA indefinitely at these tensions. In contrast, at higher tensions, (3–8 pN), we observed multiple steps upon SSB binding, with dynamic transitions among 2 to 3 distinct states (*Figure 2B*, dashed lines) depending on tension, but no dissociation of SSB. We interpret these dynamic changes in extension as wrapping and unwrapping transitions between intermediate conformations of a single ssDNA-SSB complex. Working at low SSB concentrations (0.5 nM) favored the likelihood that multiple SSBs do not bind during one cycle. We corroborated this interpretation with measurements of fluorescently labeled $SSB_f$. *Figure 2—figure supplement 1* shows that a single SSB tetramer was responsible for the observed wrapping-unwrapping dynamics. Near the dissociation force (9–10 pN), we observed multiple instances of one-step wrapping followed by complete release of ssDNA. At these forces, SSB is unable to bind the DNA tether stably, and the observed transitions correspond to protein binding and dissociation. This interpretation is also confirmed by measurements using fluorescent $SSB_f$ (*Figure 2—figure supplement 1*, right panel), in which dissociation events correlate with loss of fluorescence.

*Figure 2C* shows the combined extension change distributions from many individual SSBs at different tensions. Similarly to the force-ramp results, $\Delta x$ decreases as tension increases, indicating that the amount of ssDNA wrapped by SSB decreases. However, in contrast to the force-ramp experiment, the constant force experiment provides evidence for intermediate wrapping conformations of SSB, since multiple states are observed at many tensions. The areas under the peaks in the distributions indicate that SSB spends different amounts of time in these particular states. As tension is increased, the SSB-ssDNA complex shifts to states with smaller $\Delta x$, corresponding to lower extents of ssDNA wrapping.

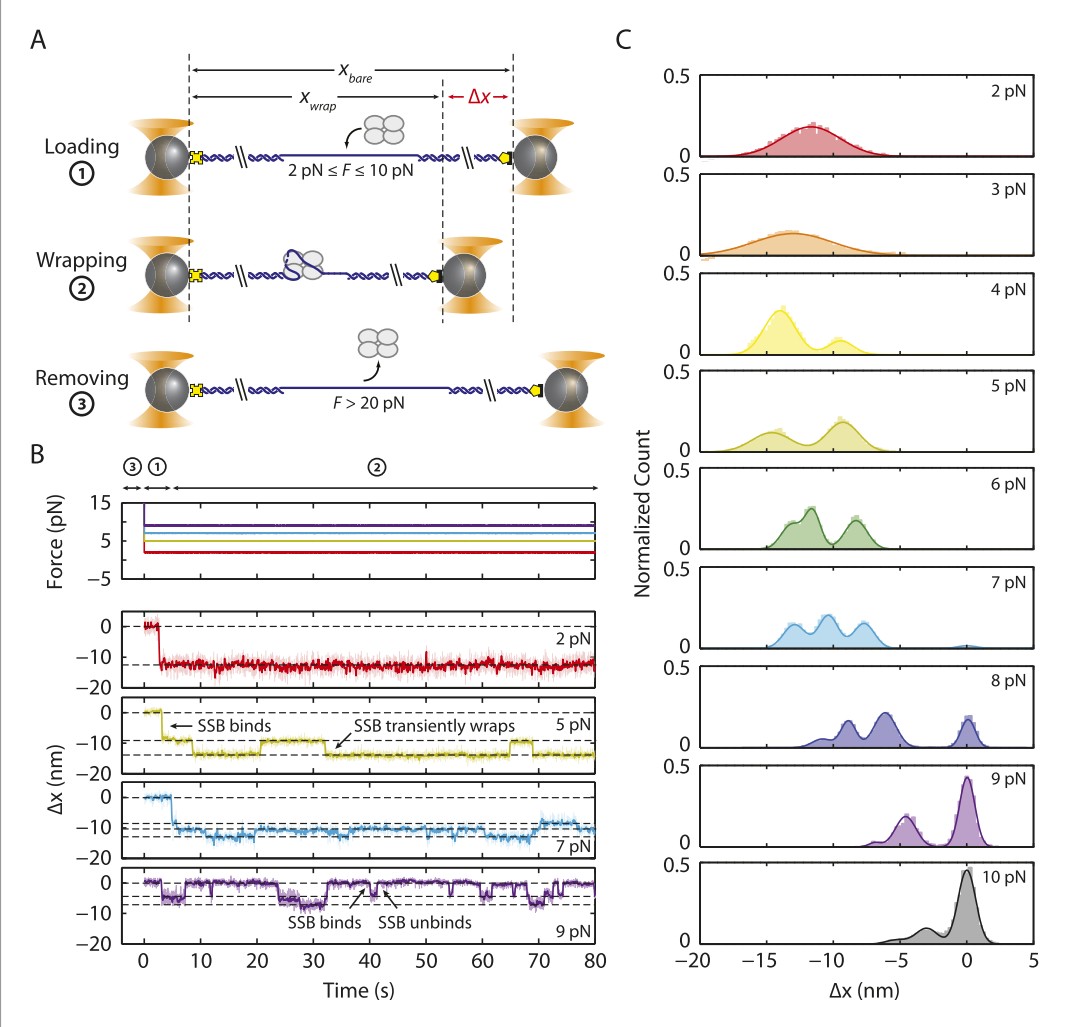

**Figure 2**. Intermediate ssDNA wrapping states of SSB under tension. (**A**) Schematic of SSB constant force wrapping experiment. A DNA construct is held between two optical traps under a constant tension between 2–10 pN in the presence of protein. An extension change, Δx, is measured upon SSB binding, wrapping or unwrapping ssDNA. At the end of each observation, SSB is removed by stretching the DNA construct to high force (>20 pN). (**B**) Representative time traces of SSB-ssDNA wrapping at 2, 5, 7, and 9 pN (red, green, blue, and purple respectively). Extension change data were acquired at 66 kHz and boxcar averaged to 10 Hz (dark color). In all traces, SSB first binds and compacts ssDNA as indicated by an extension decrease. Depending on tension, SSB displays several intermediate wrapping states. Black dashed lines represent the mean extension change of each particular wrapping state. (**C**) Extension change distribution from many SSB wrapping traces at constant tensions between 2–10 pN. The color map matches that in (**B**). Solid lines are multi-Gaussian fits to the distributions.

The following figure supplement is available for figure 2:

**Figure supplement 1**. Single SSB binding and wrapping transitions.

## Intermediates correlate with different SSB binding modes

We considered the possibility that these intermediate DNA wrapping states correspond to the different SSB binding modes observed on poly(dT) in ensemble measurements (*Bujalowski and Lohman, 1986*). *Figure 3A* displays the mean extension changes from the peaks of the distributions in *Figure 2C*. Interpreting these changes in extension, Δx, and attributing these to binding modes required a detailed model. As shown in *Figure 3B*, ssDNA wrapping by SSB contributes in two ways to the extension of the DNA tether: (i) it removes $N_w$ ssDNA nucleotides wrapped by the SSB, and (ii) it

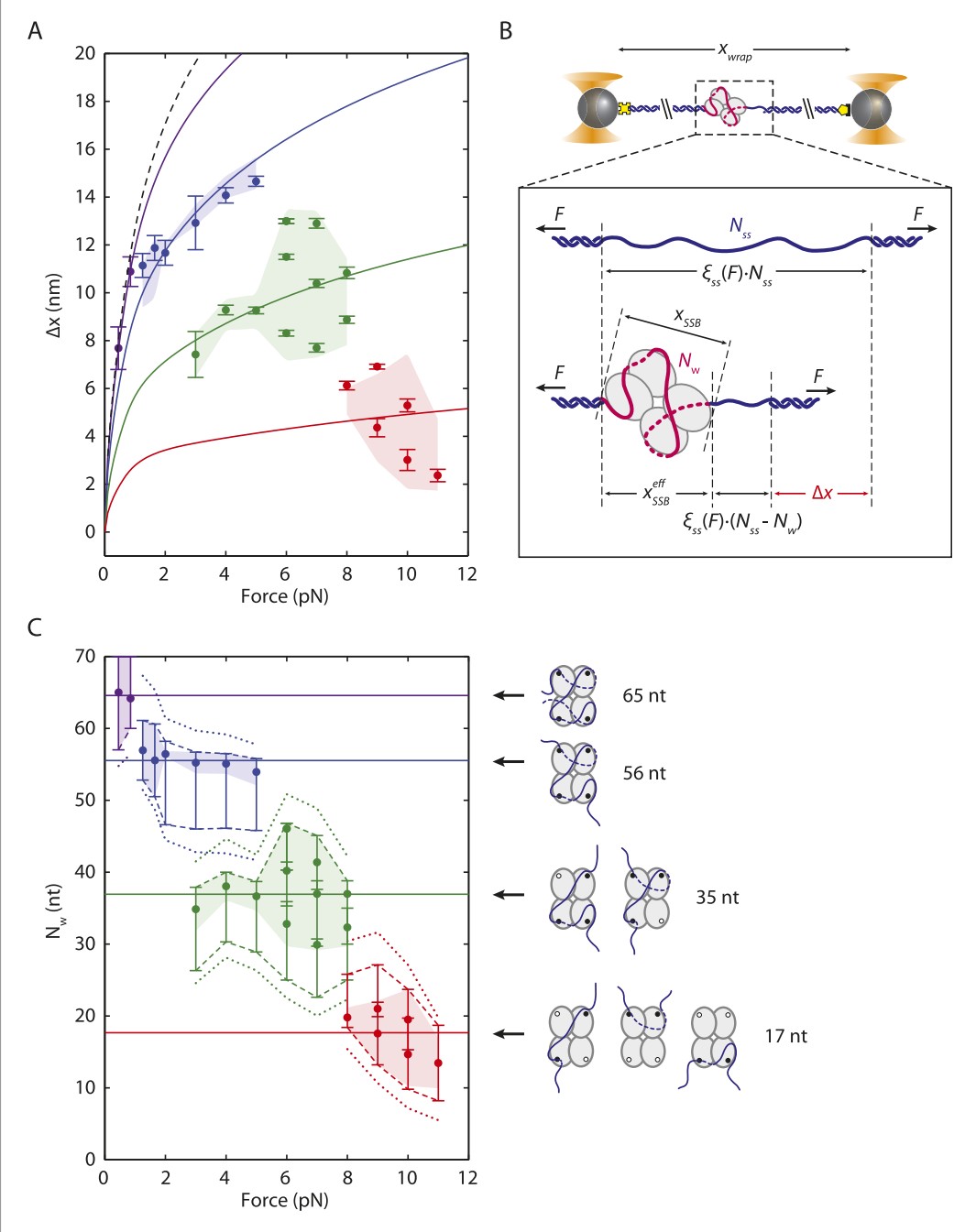

**Figure 3**. SSB wrapping modes. (**A**) Mean change in extension $\Delta x$ vs tension for each wrapping state, derived from the peaks of the distributions in **Figure 2C**. Error bars represent S.E.M. and were determined by bootstrapping. The dashed line is the model in **Figure 1D**. Solid lines represent models of $\Delta x$ based on **Equation 3** for SSB wrapping $N_w$ = 65, 56, 35, and ~17 nt (purple, blue, green, and red, respectively; 'Materials and methods'). Data points are clustered into 4 groups corresponding to those states (purple, blue, green, and red circles). (**B**) Schematic representation of $\Delta x$. Top: Bare ssDNA (with $N_{ss}$ = 70 nt) and its extension, $x_{bare}$, based on a polymer elasticity model **Equation 1** ('Materials and methods'). Bottom: SSB-wrapped ssDNA showing the number of wrapped nucleotides, $N_w$ (<70, red) and the remaining unwrapped nucleotides ($N_{ss} - N_w$, blue). The extension of wrapped DNA, $x_{wrap}$ is calculated from an elasticity model and the effective physical size of the SSB-ssDNA complex, $x_{SSB}^{eff}$, **Equation 2** ('Materials and methods'). $\Delta x$ is the difference between $x_{wrap}$ and $x_{bare}$, **Equation 3**. (**C**) Number of wrapped nucleotides $N_w$ vs tension $F$. Each data point in (**A**) is mapped to $N_w$ using the model described in the text ('Materials and methods'; **Figure 3—figure supplement 1**). Dotted lines represent the maximum possible range of

*Figure 3. continued on next page*

*Figure 3. Continued*

$N_w$ for each colored group of points based on $x_{SSB}^{eff}$ being <6.5 nm (*Figure 3—figure supplement 1*, left panel). Dashed lines represent a tighter range of possible $N_w$ for each group of points derived from the SSB-ssDNA structure (*Figure 3—figure supplement 1*, middle panel). Error bars represent this range for each individual data point. The shaded areas represent the tightest range of possible $N_w$ for each group based on the 'hotspot' analysis described in the text (*Figure 3—figure supplement 1*, right panel). The points are the best estimates of $N_w$ from the model. The shaded areas and solid lines in (**C**) map directly to those in (**A**). Cartoon schematics depict possible wrapping modes corresponding to the 4 groups.

The following figure supplements are available for figure 3:

**Figure supplement 1**. SSB wrapping models.

**Figure supplement 2**. SSB wrapping pathway.

**Figure supplement 3**. Wrapping modes of SSB mutant.

---

adds length due to the effective physical size of the SSB-ssDNA complex, $x_{SSB}^{eff}$, as noted in other mechanical unfolding studies (*Gao et al., 2012*). The extension of the wrapped DNA molecule, $x_{wrap}$, is thus:

$$x_{wrap}(F) = \xi_{ds}(F) \cdot N_{ds} + \xi_{ss}(F) \cdot (N_{ss} - N_w) + x_{SSB}^{eff}(N_w, F). \qquad (2)$$

The extension change upon wrapping, $\Delta x$, is the difference between $x_{wrap}$ and the extension of the bare molecule $x_{bare}$, given by *Equation 1*:

$$\Delta x(F) = \xi_{ss}(F) \cdot N_w - x_{SSB}^{eff}(N_{w,F}). \qquad (3)$$

$x_{SSB}^{eff}$ accounts for the distance between the two ends of the wrapped ssDNA on the SSB (*Figure 3B*). This geometrical term depends on the size of the SSB and the geometry of wrapped ssDNA around the protein, and is thus a function of $N_w$ (and $F$). For example, based on the proposed model for the (SSB)$_{65}$ structure (*Raghunathan et al., 2000*) $x_{SSB}^{eff}(N_w = 65)$ <2 nm since the ends of the wrapped ssDNA exit at nearly the same point on the protein (*Figure 1A*). In the (SSB)$_{35}$ structural model, however, the ssDNA strand exits at opposite ends of the protein and $x_{SSB}^{eff}(N_w = 35)$ is predicted to be ~5.5 nm. $x_{SSB}^{eff}$ must also account for the rotational degree of freedom of the nucleoprotein complex, and only the projection along the direction of the applied force contributes to the extension of the DNA tether. As force $F$ is exerted, a torque is applied on the complex, orienting it along the direction of tension. This effect is modeled by

$$x_{SSB}^{eff}(N_w, F) = x_{SSB}(N_w) \cdot L(Fx_{SSB}/k_B T), \qquad (4)$$

where $x_{SSB}$ is the distance between wrapped ssDNA ends in the protein's frame of reference (*Figure 3B*) and $L(z) \equiv coth(z) - 1/z$ is the orientation factor, derived from the alignment of a particle undergoing rotational Brownian motion to an external torque ('Materials and methods').

Substituting *Equation 4* into *Equation 3* provides an expression for the measured extension change $\Delta x$ at each force $F$ in terms of the SSB-ssDNA configuration parameters $N_w$ and $x_{SSB}$. Thus, for each data point $\Delta x(F)$ in *Figure 3A* there exists a set of possible values for the pair $N_w$ and $x_{SSB}$ ('Materials and methods'). *Figure 3—figure supplement 1* displays how selected data points from *Figure 3A* each project onto a curve of allowed values in the space of $N_w$ and $x_{SSB}$ (colored lines). Structural considerations limit the range of possible $N_w$ and $x_{SSB}$. The fact that $x_{SSB}^{eff}$ can be no greater than the size of the SSB (i.e., $0 < x_{SSB} < 6.5$ nm) places a restriction on the range of possible values $N_w$ can have for each $\Delta x$ (*Figure 3—figure supplement 1* left panel, dotted colored lines; *Figure 3C* dotted colored lines). We limited the range of $N_w$ further by utilizing the (SSB)$_{65}$ structure (*Raghunathan et al., 2000*) to restrict the potential geometries of any intermediate wrapping states. By measuring the end-to-end distance between every pair of nucleotides separated by $N_w$ nt along the ssDNA in the structural model, we imposed a lower and upper bound on $x_{SSB}$ at each force $F$

(*Figure 3—figure supplement 1* middle panel, gray contours and shaded area; 'Materials and methods'). This refined range of possible $N_w$ restricts our observed wrapping intermediates to four bands centered around $N_w = \sim 65$, 50–60, 30–40, and 10–20 nt (*Figure 3C* dashed colored lines). The first three correspond well with the $(SSB)_{65}$, $(SSB)_{56}$, and $(SSB)_{35}$ wrapping states observed at 25°C on poly(dT).

A better estimate for $x_{SSB}$ and $N_w$ at each force $F$ was obtained by recognizing that specific amino acid residues within *Eco*SSB are known to contact the ssDNA. Trp-40, Trp-54, Trp-88 and Phe-60 have been shown to play important roles in maintaining protein-DNA stability (*Casas-Finet et al., 1987*; *Khamis et al., 1987*; *Ferrari et al., 1997*). Crystal structure analysis also implicates Trp-54 and Arg-56 as important in creating pockets of positive electrostatic potential on the SSB surface for ssDNA to bind (*Raghunathan et al., 2000*). Lastly, a DNA density map generated by all-atom molecular dynamics (MDs) simulations of SSB (*Maffeo, 2015*) in solution with free oligonucleotides showed that DNA interacts most strongly to regions on each monomer near residues 54–56 (Trp-88 and Phe-60 are also located near this region) (*Figure 3—figure supplement 1* right schematic, residues highlighted in green; 'Materials and methods'). Based on these results, we identified the Trp-54/His-55/Arg-56 cluster as a 'hotspot', residues on each SSB monomer that may serve as anchor points along the DNA wrapping path on the SSB. Our best estimates for $N_w$ at each force $F$, shown in *Figure 3C* (colored points), were obtained by considering the distances between groups of nucleotides near each hotspot (*Figure 3—figure supplement 1* right panel, black contours; 'Materials and methods').

Our models consistently show that ssDNA unwraps in discrete steps with tension, instead of gradually as proposed previously (*Zhou et al., 2011*). As tension increases from 0–8 pN, the number of wrapped nucleotides decreases in a stepwise manner from 65 to 56 to ~35 nt (*Figure 3C*, purple, blue, and green points, respectively), matching very well to the known binding modes. The best estimates for $N_w$ and $x_{SSB}$ also generate models for the ssDNA wrapping conformations for each intermediate (*Figure 3C*; schematics and *Figure 3—figure supplement 2*). Control experiments using an SSB mutant confirm our analysis. Mutation of Trp-54 to Ser was previously shown to disrupt interactions with ssDNA and favor wrapping in the $(SSB)_{35}$ mode (*Ferrari et al., 1997*). We similarly found that the number of nucleotides wrapped by this mutant was lower than that of the wild type SSB, with $N_w = 35$ nt being the most probable wrapping conformation over the range of tensions assayed (*Figure 3—figure supplement 3*).

## SSB in intermediate wrapping states can diffuse on ssDNA

We next investigated whether the different wrapping states of SSB affect its dynamics on ssDNA, in particular its ability to diffuse. We monitored simultaneously the wrapping state of SSB and its position on ssDNA using the combined optical tweezers-confocal fluorescence microscope. We measured the latter using smFRET between the DNA construct modified with a single acceptor fluorophore (Cy5) at the 5′ ss-dsDNA junction and fluorescent $SSB_f$ labeled with an average of one donor fluorophore (AlexaFluor555) (*Figure 4A*).

Upon $SSB_f$ binding to ssDNA held at a constant 5 pN tension, we observed transitions between the two wrapping states with $N_w = 35$ nt and 56 nt, based on the analysis from the previous section. We also observed transitions between two FRET states with high ($E \sim 0.5$) and low FRET efficiencies ($E \sim 0$) corresponding to $SSB_f$ positioned at the 5′ ss-dsDNA junction vs the 3′ end, respectively. As shown in *Figure 4B*, all four combined extension-FRET states could be detected in our data: 'i'—35 nt wrapping and low FRET, 'ii'—35 nt wrapping and high FRET, 'iii'—56 nt wrapping and high FRET, and 'iv'—56 nt wrapping and low FRET. Inspection of individual time traces revealed cases in which transitions in extension and FRET were correlated. *Figure 4C* (left) shows an example of such a transition from state i → iii → i, in which an SSB in $(SSB)_{35}$ mode wraps an additional ~20 nt of ssDNA from the 5′ end into $(SSB)_{56}$ mode, then releases the same end of DNA. This confirms our interpretation that these changes in extension represent transitions between binding modes. Alternately (*Figure 4C*; middle and right) we observed cases in which FRET transitions occurred independently of changes in wrapping state. The two-state time traces indicate SSB diffusing across the sensitive distance range of smFRET (about one Förster radius, ~6 nm = 18 nt [*Forster, 1948*]) and support a reptation mechanism for SSB diffusion (*Figure 4—figure supplement 1*), as previously proposed (*Zhou et al., 2011*). Diffusion of SSB occurred in both $(SSB)_{35}$ (*Figure 4C*; middle) and $(SSB)_{56}$ (*Figure 4C*; right) wrapping modes. We reasoned that the lifetimes of the high FRET states in these traces correspond approximately to the

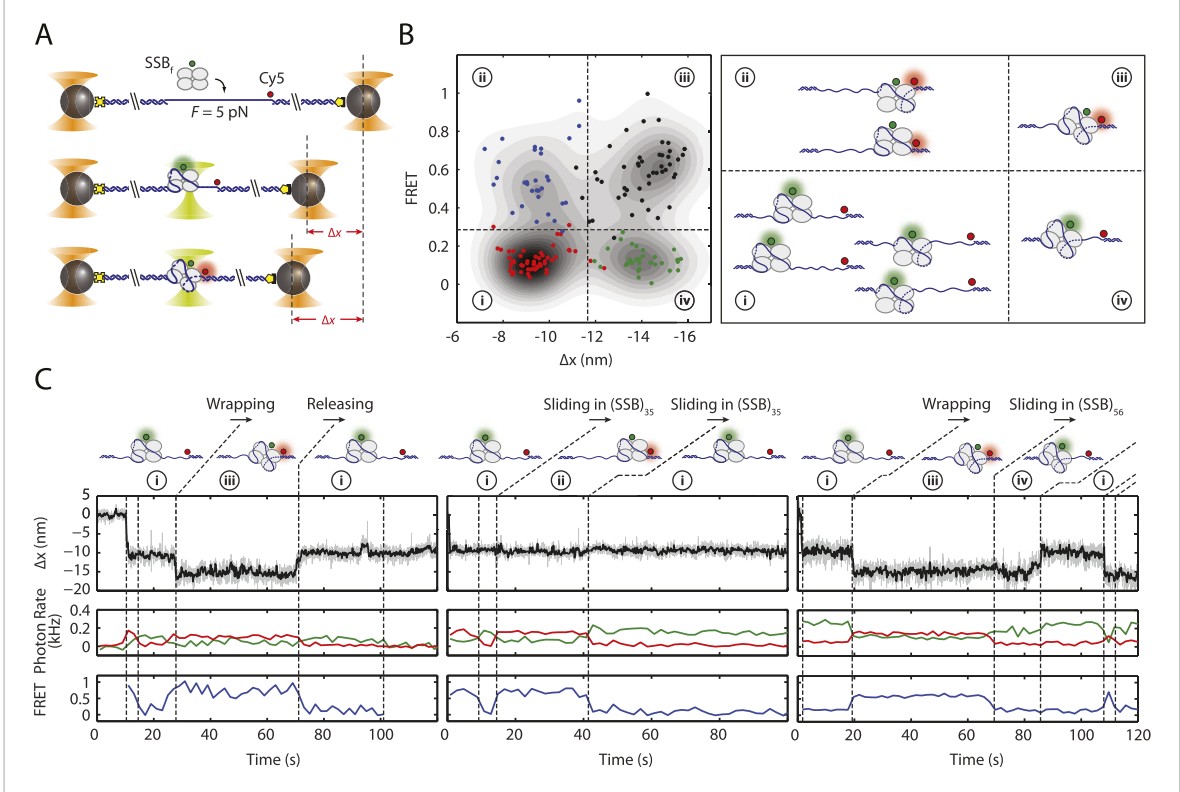

**Figure 4**. SSB binding modes and diffusion mechanism. (**A**) Schematic of fluorescently labeled SSB, $SSB_f$, ssDNA wrapping experiment. A Cy5-labeled DNA construct is tethered between two optical traps under a constant tension of 5 pN. Upon binding of an AlexaFluor555-labeled SSB, both DNA extension change, $\Delta x$, and single-molecule FRET are measured simultaneously. (**B**) Scatter plot of FRET efficiency and $\Delta x$. Data (circles) are assigned to 4 states (red (i), blue (ii), black (iii), and green (iv)) based on the value of FRET and $\Delta x$. A density map of the combined FRET-extension states overlaid with the scatter plot confirms that the data can be separated into 4 states. Cartoon illustrations of nucleoprotein complexes demonstrate possible SSB wrapping configurations corresponding to the 4 assigned states. (**C**) Representative traces showing combined fluorescence and DNA extension measurements. Change in extension (top; boxcar averaged to 50 Hz) and fluorescence (middle; boxcar averaged to 0.5 Hz) of donor ($SSB_f$, green) and acceptor (Cy5, red) are measured simultaneously. Together, FRET efficiency (bottom; blue) and extension change (top; black) reveal the SSB wrapping states (i and ii, iii and iv) and their dynamics (ssDNA wrapping/releasing and sliding).

The following figure supplement is available for figure 4:

**Figure supplement 1**. Mechanism of SSB diffusion.

time the protein takes to diffuse by one Förster radius from the ss-dsDNA junction, and estimated a diffusion constant $D \approx 27$ $nt^2/s$ for the $(SSB)_{35}$ mode and 15 $nt^2/s$ for the $(SSB)_{56}$ mode. This range of values is consistent with prior reports (*Roy et al., 2009*) when accounting for temperature (~23°C in our measurements) and the expected reduction in $D$ due to the 5 pN tension (*Roy et al., 2009*; *Zhou et al., 2011*). We observed no examples (0 of $N = 82$) of transitions from state i → iii → ii—wrapping one end of DNA and releasing the other—providing no support for a 'rolling' mechanism of diffusion (*Romer et al., 1984*) (*Figure 4—figure supplement 1*).

## Discussion

Due to its homotetrameric nature, the *Eco*SSB protein can bind ssDNA in a number of different modes that differ in the number of nucleotides occluded in complexes with long ssDNA (*Bujalowski and Lohman, 1986*; *Lohman and Ferrari, 1994*; *Roy et al., 2007*). SSB-ssDNA complexes can transition between these modes in vitro and their stabilities can be modulated by changes in solution conditions (salt, pH, temperature) as well as the SSB to DNA ratio. Our experiments show that force can also be used to control the ssDNA wrapping state of *Eco*SSB. This has revealed stable intermediate states of

$(dT)_{70}$ ssDNA wrapping around a single SSB tetramer that correlate well with the known [NaCl]-induced poly(dT) binding modes, $(SSB)_{65}$, $(SSB)_{56}$, $(SSB)_{35}$ that have been observed for SSB binding to longer poly(dT) (*Lohman and Overman, 1985*; *Bujalowski and Lohman, 1986*).

The observation of stable force-induced SSB-$(dT)_{70}$ intermediates provides new details about the likely wrapping topologies of the different binding modes. Our results are consistent with the ssDNA wrapping topology proposed for the $(SSB)_{65}$ mode based on a crystal structure (*Figure 3C*; schematic, and *Figure 3—figure supplement 2*) (*Raghunathan et al., 2000*). They also suggest that the $(SSB)_{56}$ mode has ssDNA bound to all four subunits, but with the 3′ terminal ssDNA end unraveled to the nearest hotspot (*Figure 3C*; schematic, and *Figure 3—figure supplement 2*). This model is consistent with studies (*Bujalowski and Lohman, 1989a*, *1989b*) suggesting that all 4 monomers of an SSB tetramer interact with ssDNA upon binding a molecule of $(dT)_{56}$. At forces in the range of 5–8 pN, we observe between 1 to 3 separate states wrapping 30–40 nt. Our data and analysis are not sensitive enough to ascribe specific wrapping conformations to each. We believe at least two conformations wrapping ∼35 nt are consistent with the observed extension changes, one of which is nearly identical to the proposed $(SSB)_{35}$ structure (*Raghunathan et al., 2000*) (*Figure 3C* schematic, and *Figure 3—figure supplement 2*). Interestingly, prior studies (*Roy et al., 2007*) have suggested the existence of an alternate '$(SSB)_{35b}$' mode that occludes 35 nt but is structurally distinct from $(SSB)_{35}$, consistent with our observations. At tensions >8 pN, we also observed a stable intermediate reflecting ∼17 nt of bound ssDNA (*Bujalowski and Lohman, 1989a*, *1989b*, *1991b*). Here, a multitude of wrapping conformations around two monomers is consistent with the data (*Figure 3C* schematic, and *Figure 3—figure supplement 2*). Although fluorescence quenching studies (*Bujalowski and Lohman, 1991a*) suggest that $(dT)_{16}$ would bind to one monomer of SSB, partial interactions with two monomers in our structural model may sum to those of a monomer. It is possible that near dissociation, wrapping geometries could be more heterogeneous. Prior studies have shown that *Eco*SSB can bind to ssDNA as short as $(dT)_8$ (*Krauss et al., 1981*). However, we do not observe long-lived intermediates wrapping less than ∼17 nt before SSB dissociation.

Analyzing the transitions between wrapping intermediates (*Figure 2B*) reveals that almost every transition ($N$ = 373 out of 380 total, 98%) occurs between adjacent wrapping states, that is, between $(SSB)_{56}$ and $(SSB)_{35}$, but never directly between $(SSB)_{56}$ and $(SSB)_{17}$. This suggests a single, linear kinetic pathway for wrapping (*Figure 3—figure supplement 2*, right to left) and unwrapping (left to right). This proposed pathway is corroborated by measurements of *E. coli* SSB in competition with RecA for ssDNA. As shown in *Figure 5A,B*, we first loaded a single SSB tetramer onto ssDNA at a force of 5 pN, where our analysis shows the protein interconverts between the $(SSB)_{56}$ and $(SSB)_{35}$ modes. We then added RecA to the complex under conditions favoring polymerization into ssDNA-RecA filaments ('Materials and methods'). (To prevent polymerization of RecA onto the dsDNA handles, the construct was synthesized with the 70-nt ssDNA loading site flanked by short non-DNA spacers ['Materials and methods']). In the absence of SSB, RecA extends the construct by ∼10 nm as it fills the ssDNA (*Figure 5—figure supplement 1*), consistent with previous reports that ssDNA-RecA filaments are 50% longer than dsDNA (*Hegner et al., 1999*; *Galletto et al., 2006*) ('Materials and methods'). When RecA is added to ssDNA wrapped by a single SSB, RecA takes longer to polymerize but eventually removes the SSB in a stepwise fashion (*Figure 5C*). Analyzing the measured extension changes from many measurements (*Figure 5D*; 'Materials and methods') reveals that the SSB is unraveled in discrete steps, corresponding to the same pathway of intermediates, $(SSB)_{35} \rightarrow (SSB)_{17} \rightarrow$ unbound, as proposed above (*Figure 3—figure supplement 2*).

The ability to measure the extension of each wrapping state as a function of force also allows us to construct an energy landscape for the SSB-ssDNA complex. Using the extension histograms in *Figure 2C*, we determined the probabilities of occupying specific wrapping modes at each force, and from these we calculated the free energy differences between modes ('Materials and methods'; for simplicity, we ascribed intermediates with similar $N_w$ to the same wrapping state). We also used the lifetimes of each wrapping state and transition probabilities at each force (*Figure 2B*) to estimate the barrier heights between states ('Materials and methods'). Our analysis (*Figure 6*) shows that the free energy of wrapping into the $(SSB)_{65}$ mode is 21 ± 1 $k_BT$, in excellent agreement with the area between protein-bound and bare FECs (22 ± 2 $k_BT$; *Figure 1D*). Interestingly, this wrapping free energy is not distributed evenly among the 65 nt. Instead, we find that 73% of the energy is concentrated in the first 35 nt wrapped (energy density = 0.44 ± 0.02 $k_BT$/nt). In contrast, the $(SSB)_{65}$ and $(SSB)_{56}$ states are separated by only ∼0.7 $k_BT$ (energy density ∼0.07 $k_BT$/nt). This finding suggests that the last ∼10 nt

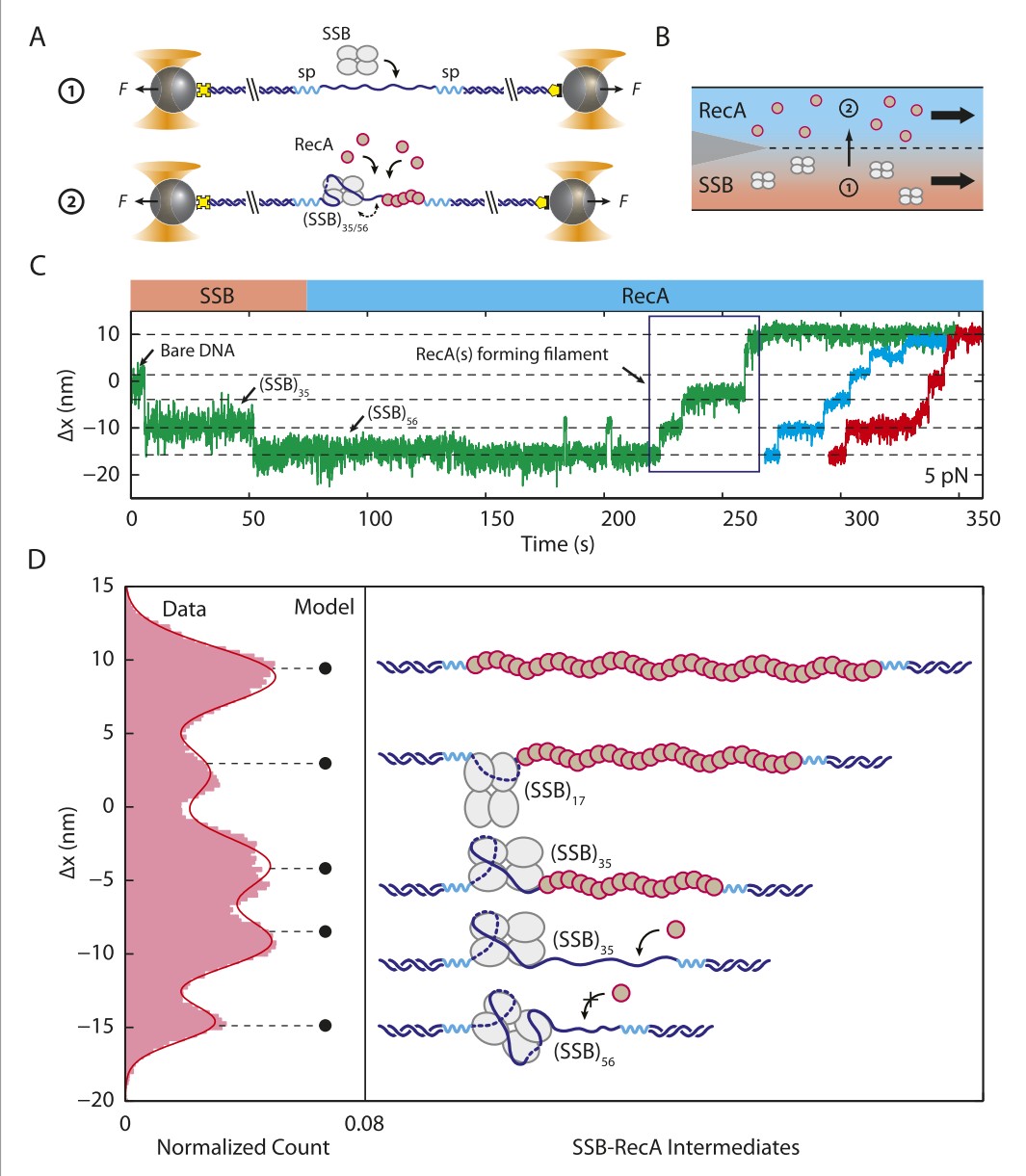

**Figure 5**. Unwrapping of ssDNA from SSB by RecA filament formation. (**A**) Schematic representation of SSB-RecA experiment. A standard DNA construct consisting of a 70-nt single-stranded DNA ((dT)$_{70}$) fragment was synthesized to contain two internal 18-atom hexa-ethylene-glycol spacers at both ss-dsDNA junctions (cyan; 'Materials and methods'). The spacers prevent RecA filament formation onto the dsDNA. The construct is tethered in the presence of SSB (Position 1). After the SSB binds, the tethered DNA is moved to the stream containing RecA for observation (Position 2). (**B**) Experimental flow chamber for SSB-RecA experiment. Two separate streams contain experimental buffer plus 0.5 nM SSB (red, Position 1) and buffer plus 125 nM RecA and 125 µM ATP-γS (blue, Position 2). (**C**) Representative time traces showing competition between RecA and SSB on ssDNA (green, blue, red). Transient wrapping-unwrapping of SSB slows down the nucleation of RecA. Formation of RecA filament extends ssDNA (blue box), displaces the SSB, and stops after reaching the spacers at the ss-dsDNA junctions. The dotted lines correspond to the model in (**D**). (**D**) Extension change distribution of SSB-RecA intermediates at a constant tension of 5 pN (pink) obtained from many RecA filament formation time traces (*N* = 25). Five states representing SSB-RecA dissociation intermediates are illustrated (schematics) and assigned to peaks of the distribution. Extensions corresponding to these states are predicted using polymer models of elasticity (black dots and dotted lines, 'Materials and methods').

The following figure supplement is available for figure 5:

**Figure supplement 1**. RecA filament formation on modified single-stranded DNA.

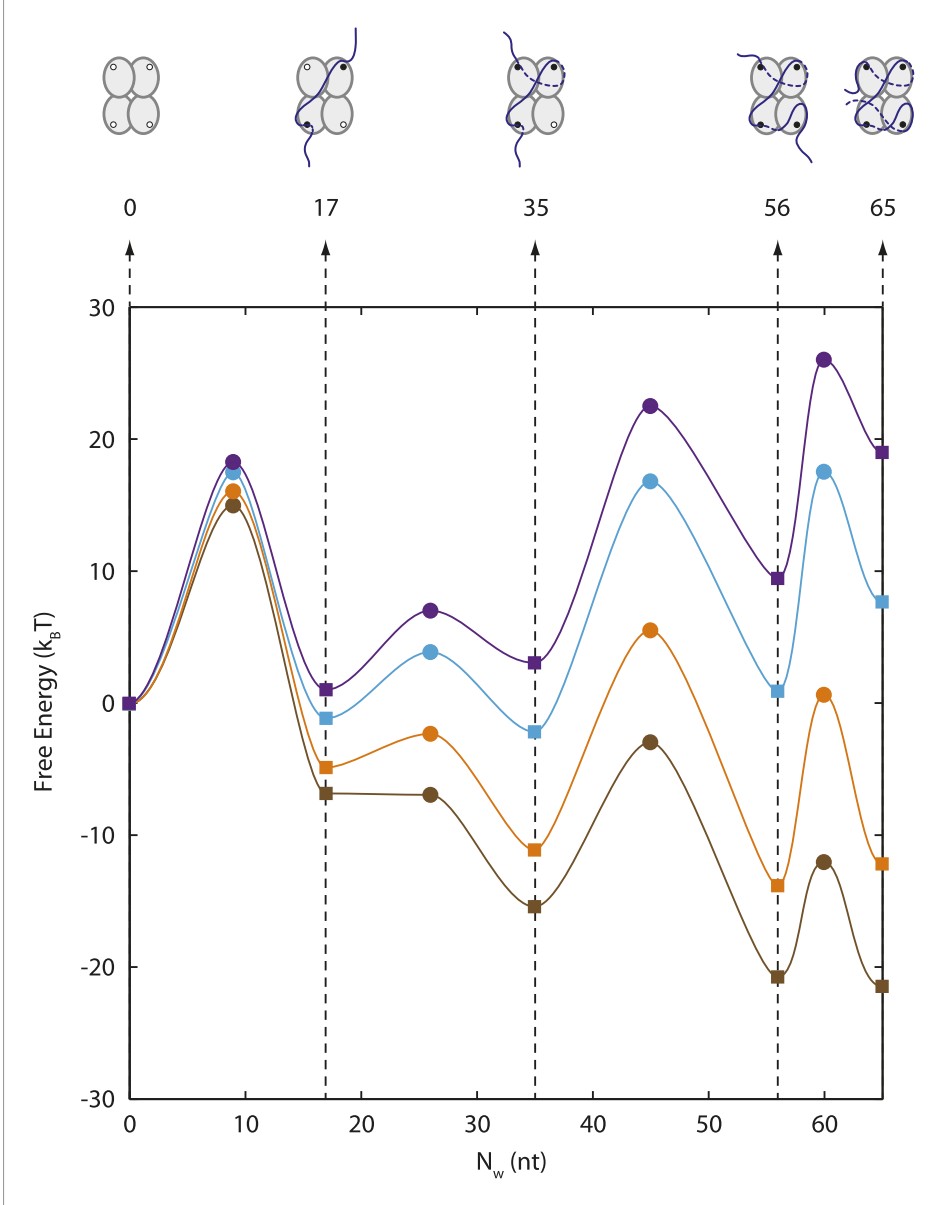

**Figure 6**. Energy landscape of SSB wrapping. Energy landscapes of a single SSB wrapping ssDNA at representative forces reconstructed from extension change probability distributions vs tension (*Figure 2C*). The potential wells correspond to the stable SSB-ssDNA intermediates (cartoon schematics): $(SSB)_{65}$, $(SSB)_{56}$, $(SSB)_{35}$, $(SSB)_{17}$, and unbound, respectively. The energy associated with each intermediate is determined from the occurrence probabilities for each state (squares, 'Materials and methods'). The barrier heights and positions (circles) are determined from the state lifetimes ('Materials and methods'). In the absence of tension, SSB wraps ssDNA in the $(SSB)_{65}$ binding mode. Increasing tension (brown, orange, cyan, purple lines correspond to 0, 3, 7, 9 pN, respectively) tilts the energy landscape, changes the free-energy difference between wrapping intermediates, and favors different SSB-ssDNA binding modes.

The following figure supplements are available for figure 6:

**Figure supplement 1**. Occurrence probability of SSB wrapping intermediates.

**Figure supplement 2**. Modeling of transition rates between SSB wrapping intermediates.

wrapped are more susceptible to unraveling and thus might be more accessible to other proteins competing for ssDNA. This unbalanced energy density profile may provide a mechanism by which SSB is displaced by the recombinase RecA, which requires a foothold of 6–17 nt to polymerize into filaments (*Joo et al., 2006*; *Bell et al., 2012*). We note that in the RecA/SSB competition experiment (*Figure 5*), we observe RecA filaments forming only once the SSB transitions to the $(SSB)_{35}$ mode, granting access to >14 nt of ssDNA.

Our measurements that SSB can diffuse on ssDNA while in different wrapping modes provide insights into how SSBs could be redistributed along ssDNA by other proteins seeking access to ssDNA. The observation of SSB-ssDNA rearrangements without unwrapping or rewrapping (*Figure 4*) points to a sliding mechanism of diffusion in which ssDNA reptates along the protein, consistent with prior models (*Zhou et al., 2011*). In *Figure 5*, we believe RecA polymerization likely slides the SSB to one ssDNA-dsDNA junction prior to unravelling it (*Roy et al., 2009*; *Bell et al., 2012*). Interestingly, the data in *Figure 4* suggest that diffusion may be faster in the $(SSB)_{35}$ mode. The transition rates between FRET states are $\sim 1.8\times$ larger in the $(SSB)_{35}$ mode than in the $(SSB)_{56}$ mode. The observation that a smaller site size leads to faster diffusion is consistent with reports that human RPA, which covers 30 nt, has a larger diffusion coefficient than *Eco*SSB in its $(SSB)_{65}$ mode (*Nguyen et al., 2014*).

Previous work has proposed that different wrapping modes may be used selectively in different DNA metabolic processes (e.g., replication vs recombination) (*Sancar et al., 1981*; *Lohman et al., 1988*). How and which of these modes are used for particular processes remains unclear, as experimental proof of this proposition has proven difficult to obtain in vitro. We anticipate that the control of SSB wrapping mode by applied force may be a useful experimental tool to test this hypothesis.

## Materials and methods

### Sample preparation
#### SSB, fluorescently labeled SSB, and RecA
Both wild-type and fluorescently labeled *E. coli* SSB were expressed and purified as described previously (*Lohman et al., 1986*; *Roy et al., 2009*), with an addition of a dsDNA cellulose column to remove a minor exonuclease contaminant (*Bujalowski and Lohman, 1991b*). The labeled SSB was single-point mutated from Ala to Cys at position 122 in the C-terminus, and labeled with AlexaFluor555 maleimide (Invitrogen, Grand Island, NY) to the extent of $\sim 25\%$ ($\sim 1$ dye per tetramer) as described previously (*Roy et al., 2009*). *E. coli* RecA was purchased from New England Biolabs (M0249S; Ipswich, MA).

#### Single-stranded DNA construct
The single-stranded DNA construct consisted of three separate fragments ligated together (*Figure 1—figure supplement 5*): 'Right Handle' (RH), 'Left Handle' (LH), and 'Binding Site' (BS). The handles served as functionalized linkers that connected to trapped beads through biotin-streptavidin and digoxigenin-anti-digoxigenin linkages and spatially separated the beads from the protein binding site. LH was synthesized from PCR amplification of the PBR322 plasmid (New England Biolabs, Ipswich, MA) using a 5′-biotin-labeled primer and digested to a 1550-bp length with the PspGI restriction enzyme (New England Biolabs, Ipswich, MA), leaving a 5-nt 5′ overhang. RH was PCR-amplified from the phage lambda DNA (New England Biolabs, Ipswich, MA) using a 5′-digoxigenin-labeled primer and digested with the TspRI restriction enzyme (New England Biolabs, Ipswich, MA), resulting in a 1710-bp dsDNA with a 9-nt 3′ overhang.

The last fragment of the construct, BS, consisted of a 70-nt poly(dT) oligodeoxyribonucleotide flanked by sequences complementary to both overhangs of LH and RH: 5′-CCTGG $(T)_{70}$ CCCACTGGC-3′. In some experiments, a Cy5 fluorescence dye was attached directly to the DNA backbone using phosphoramidite chemistry at the location between the 5′ complementary sequence and the 70-nt poly(dT) region. The final construct had one digoxigenin and one biotin on opposing ends for linkages to anti-digoxigenin- and streptavidin-coated beads, respectively. All oligonucleotides were custom-ordered from Integrated DNA Technologies (Coralville, IA).

In the experiments with RecA, BS was modified to contain two internal 18-atom hexa-ethylene-glycol spacers (iSp18; Integrated DNA Technology, Coralville, IA) between the 70-nt poly(dT) and the complementary overhangs. This modification prevented RecA filament formation onto the dsDNA handles (*Figure 5A*, cyan). The BS fragment was ligated to RH and LH to form a complete construct.

## Instrument design

### Optical tweezers

Experiments were performed using a high-resolution dual optical trap instrument combined with a confocal microscope as previously described (*Comstock et al., 2011*). The dual traps were formed by timesharing a single IR laser (a 5-W, 1064-nm diode-pumped solid-state laser, YLR-5-1064-LP; IPG Photonics, Oxford, MA), by intermittently deflecting the laser between two angles with an acousto-optic modulator (AOM; IntraAction Corp., Bellwood, IL). The instrument was housed in a temperature-controlled room at ~23°C.

The IR beams were tightly focused by a 60×, water-immersion microscope objective (Nikon, Tokyo, Japan) to form two optical traps inside the sample chamber. Each trap held a single polystyrene bead during an experiment. Bead displacements were detected by back–focal plane interferometry: forward-scattered laser light was collected by a second identical objective lens, imaged onto a quadrant photodiode detector, and analyzed. In all experiments, both traps were calibrated by measuring the power spectral density of bead Brownian motion. Trap stiffnesses were typically equal to 0.3 pN/nm.

Fluorescence probes were excited by a 532-nm 5-mW laser (DPGL-05S, World Star Tech, Toronto, ON, Canada) interlaced with the trapping IR laser at a rate of 66 kHz (*Comstock et al., 2011*). Fluorescence light from donor and acceptor dyes emitted from within a confocal volume was collected by the front objective, band-pass filtered, focused through a 20-μm pinhole, and imaged onto two avalanche photodiodes (APDs) (PerkinElmer, Waltham, MA). The AlexaFluor555 emission passed through a 580-nm low-pass filter (Chroma Technology Corp., Bellows Falls, VT) to one APD, and the Cy5 emission through a 680-nm low-pass filter to the second APD.

### Flow chamber

A custom-designed laminar flow chamber ([*Brewer and Bianco, 2008*], *Figure 1—figure supplement 4*), consisting of two glass coverslips (12-545-M, 24 × 60-1, ThermoFisher, Waltham, Massachusetts) sandwiching melted Nescofilm (Karlan, Phoenix, AZ) was patterned with channels. Eight holes with a diameter of 2 mm were drilled onto one of the coverslips by a laser engraver system (VLS2.30; Universal Laser Systems, Scottsdale, AZ) to create four inlets and four outlets. The Nescofilm was cut into three separate channels using the same laser system. Top and bottom channels were connected to a central channel through glass capillaries (OD = 100 ± 10 μm, ID = 25.0 ± 6.4 μm; Garner Glass Co., Claremont, CA). The chamber was mounted onto an anodized aluminum frame into which inlet and outlet tubing (ABW00001; Tygon, Saint-Gobain, Akron, OH and PE20; Intramedic, Becton Dickinson and Company, Sparks, MD) was connected.

Three syringe pumps (PHD 2000 Infusion; Harvard Apparatus, Holliston, MA) were used to control the flow through the different channels: top, central, and bottom, separately (*Figure 1—figure supplement 4*). The top and bottom channels were injected with anti-digoxigenin and streptavidin beads, respectively. In the central channel, two streams of appropriate buffers were pumped at a speed of 140 μm/s (~100 μl/hr) and merged to form a laminar interface. In a typical experiment, a DNA molecule tethered between trapped beads could be moved across the interface using a motorized stage controller in ~2 s.

### Optical tweezers experiment

Except where otherwise noted, experiments were performed in a working buffer containing 100 mM Tris-HCl (pH 7.6), 10 mM NaCl, 0.1 mM EDTA. An oxygen scavenging system (pyranose oxidase [P4234; Sigma–Aldrich, St. Louis, MO] and catalase [219001; EMD Millipore, Billerica, MA]) was added to increase tether and fluorescence photobleaching lifetime (*Landry et al., 2009*); to this buffer, 0.5 nM of SSB protein was added. For the measurements involving fluorescence, an oxygen triplet-state quencher (Trolox; Sigma–Aldrich, St. Louis, MO) was added to the working buffer to prevent fluorophore blinking (*Rasnik et al., 2006*). Experimental conditions were chosen to be compatible with the optical trapping assay and to favor the $(SSB)_{65}$ mode in the absence of force. The $(SSB)_{65}$ mode is known to be stabilized at high [NaCl] (>200 mM), the $(SSB)_{56}$ mode at intermediate [NaCl] (50–100 mM), and the $(SSB)_{35}$ mode at low [NaCl] (10 mM) (*Bujalowski and Lohman, 1986*). $Mg^{2+}$ and polyamines also facilitate formation of the high site size modes (*Bujalowski and Lohman, 1986*; *Wei et al., 1992*). We independently verified that the $(SSB)_{65}$ mode was favored in the experimental conditions above (100 mM Tris-HCl, low SSB concentration), by measuring a binding isotherm using fluorescence of $Cy5-(dT)_{70}-Cy3-dT$ with SSB (*Figure 1—figure supplement 6*).

In all experiments, a single-stranded DNA construct was first tethered between a trapped streptavidin-coated bead and an anti-digoxigenin-coated bead in buffer. The tether was then stretched under tension to obtain a FEC. The FEC was used to check behavior of the tether by verifying it against a theoretical polymer model (*Figure 1—figure supplement 1*).

## Force-ramp experiment

A tether was moved into the SSB stream at low tension to allow a single SSB to bind (*Figure 1—figure supplement 4*, Position 2). After a period of incubation, the tether was moved back to the blank buffer (Position 1) to ensure that no other SSBs were present during experiment. To observe single SSB unwrapping, a force-ramp experiment was performed by increasing the trap separation at a rate of ~65 nm/s until the tether tension reached ~25 pN. The tether was then relaxed back at the same rate to the original starting position.

## Constant force experiment

Constant force experiments were performed with a PID controller loop that monitored the trapped bead positions and controlled the trap separation to maintain a constant tension on a tethered DNA molecule. The constant force experiment was initiated in the blank buffer stream at constant tensions ranging from 2 to 11 pN (*Figure 1—figure supplement 4*, Position 1). While keeping tension constant, the tether was moved into the SSB stream to allow a single SSB to bind (Position 2). After an SSB bound, the tether was moved back to the blank buffer stream for observation.

## RecA-SSB competition experiment

These experiments were performed in a working buffer containing 20 mM Tris(OAc), pH 7.5, 10 mM NaCl, 4 mM Mg(OAc)$_2$, and an oxygen scavenging system. The experiment was initiated in a buffer stream containing 0.5 nM of SSB only at a constant tension of 5 pN (*Figure 5A,B*). After an SSB bound (*Figure 5B*; Position 1), the tethered construct was moved into the buffer stream containing 125 nM of RecA and 125 μM ATP-γS for observation (Position 2). ATP-γS (A1388; Sigma–Aldrich, St. Louis, MO) was included to stabilize the RecA filament.

# Data analysis

## Single-stranded DNA polymer modeling

The total extension of the tether was decomposed into dsDNA and ssDNA components as shown in *Equation 1*. The extension of each of these segments was computed separately. The dsDNA segment was modeled with an XWLC (*Bustamante et al., 1994*). Parameters for dsDNA were obtained from the literature (*Baumann et al., 1997*); we used a persistence length of 53 nm, a stretch modulus of 1200 pN, and a contour length per base pair of 0.338 nm bp$^{-1}$. The ssDNA segment was fitted to the recently reported 'snake-like' chain model (*Saleh et al., 2009*). Parameters were obtained by comparing the amount of salt (monovalent ion) used in our buffer to the lookup table provided (*Saleh et al., 2009*). Representative FECs of the DNA construct containing 3260 bp dsDNA and 70 or 140-nt poly(dT) ssDNA (*Figure 1—figure supplement 2*; green and orange, respectively) were fitted to the model (black dashed and dotted lines, respectively). FEC data of both constructs were in excellent agreement with theoretical models of DNA elasticity.

We validated the use of the SLC model for ssDNA of varying lengths by subtracting FECs of a construct containing a 70-nt ssDNA site (red) from those of a construct with a 140-nt poly(dT) ssDNA site (orange) at each force. The resulting extension difference (*Figure 1—figure supplement 2*, inset) displayed an excellent agreement with the SLC model for 70 nt (black dashed line). (The extension difference was also used to determine one of the parameters of the SLC model, the ssDNA extension at 20 pN [*Saleh et al., 2009*]. For 70-nt ssDNA, this was determined to be ~35 nm.)

## SSB-ssDNA complex modeling

### Modeling the effect of SSB-ssDNA complex size on extension

*Equation 2* models the extension of SSB-wrapped DNA. The second term in the expression represents the extension due to the remaining $N_{ss} - N_w$ nucleotides of ssDNA unwrapped by the protein, and the third represents the contribution to the extension from the physical size of the SSB-ssDNA complex. For the latter, we approximated the ssDNA-wrapped SSB as a rigid body of size $x_{SSB}$

that is able to diffuse rotationally. The effect of tension $F$ on the ssDNA is to orient the complex along the direction of tension. The energy associated with orienting the SSB-ssDNA complex is given by:

$$E_{orient} = -\vec{F} \cdot \vec{x}_{SSB} = -Fx_{SSB}\cos\theta,$$

where $\vec{F}$ is the force vector, $\vec{x}_{SSB}$ the vector defined by the entry and exit points of the wrapped ssDNA on the protein (*Figure 3B*), and θ is the angle between the two vectors. The effective size of the SSB, that is, that which contributes to the measured extension, is given by the projection of $\vec{x}_{SSB}$ onto the force axis, $x_{SSB}^{eff} = x_{SSB}\langle\cos\theta\rangle$, where $\langle\ldots\rangle$ denotes the thermal average. This average is obtained by integrating a Boltzmann distribution of orientation energies over all possible orientation angles θ, φ:

$$\langle\cos\theta\rangle = \frac{\int\limits_0^{2\pi} d\varphi \int\limits_0^{\pi} \sin\theta d\theta \cos\theta \exp(-Fx_{SSB}\cos\theta/k_BT)}{\int\limits_0^{2\pi} d\varphi \int\limits_0^{\pi} \sin\theta d\theta \exp(-Fx_{SSB}\cos\theta/k_BT)}.$$

Note that θ, φ correspond to the angles in a spherical coordinate system with force pointing along the *z*-axis. Carrying out the integrals yields:

$$\langle\cos\theta\rangle = \coth\left(\frac{Fx_{SSB}}{k_BT}\right) - \frac{k_BT}{Fx_{SSB}},$$

known as the Langevin function, $L(Fx_{SSB}/k_BT)$ in *Equation 4*, first derived for the classical model of paramagnetism (*Langevin, 1905*). The same expression has also been used to model protein size effects in mechanical unfolding studies (*Chen et al., 2015*). For forces $F >> k_BT/x_{SSB}$, the complex aligns with the force vector and $\langle\cos\theta\rangle \approx 1$.

## Determination of SSB wrapping conformation from extension change data

*Equations 3, 4* relate the measured extension change $\Delta x$ at each force $F$ to the number of wrapped nucleotides, $N_w$, and the distance between ssDNA entry and exit points on the SSB, $x_{SSB}$. Substituting *Equation 4* into *Equation 3* and solving for $N_w$ yields

$$N_w = \frac{\Delta x(F) + x_{SSB}\coth(Fx_{SSB}/k_BT) - k_BT/F}{\xi_{ss}(F)}, \tag{5}$$

where the definition of the Langevin function $L(z)$ was used. Entering an extension change data point $\Delta x(F)$ and ssDNA elasticity model value $\xi_{ss}(F)$ into *Equation 5* at a given force $F$ yields a single-valued function of $N_w$ in terms of $x_{SSB}$. The functions $N_w(x_{SSB})$ represent the set of allowable values of the pair $x_{SSB}$, $N_w$ for each extension change data point $\Delta x(F)$, and are plotted as colored curves in *Figure 3—figure supplement 1* for selected data points from *Figure 3A*. The widths of the curves correspond to the error bars in *Figure 3A*.

We restricted the range of allowable values for $N_w$ by placing upper and lower limits on $x_{SSB}$, $x_{SSB,max}$ and $x_{SSB,min}$, based on structural constraints. At coarsest level, $x_{SSB}$ is bounded by the size of the protein, such that $x_{SSB,min} = 0$ and $x_{SSB,max} = 6.5$ nm. This provided upper and lower limits on $N_w$ for each data point $\Delta x(F)$ (*Figure 3—figure supplement 1* left panel, dotted colored lines). A stricter set of constraints was obtained from the maximum and minimum end-to-end distances between pairs of wrapped nucleotides $n_i$ and $n_j$ separated by $N_w$ nt (i.e., $|n_i - n_j| = N_w - 1$). We used the SSB-ssDNA crystal structure (*Raghunathan et al., 2000*) to determine these bounds, $x_{SSB,max}(N_w)$ and $x_{SSB,min}(N_w)$ (*Figure 3—figure supplement 1* middle panel, gray contours and shaded area). The intersection points between the curves generated by *Equation 5* and $x_{SSB,max}(N_w)$ and $x_{SSB,min}(N_w)$ provided a tighter set of limits on $N_w$ for each data point $\Delta x(F)$ (*Figure 3—figure supplement 1* middle panel, dashed colored lines).

The best estimates for $N_w$ were obtained by considering 'hotspots' of interactions. Clusters of residues on the SSB tetramer to which nucleotides preferentially associated were determined from the SSB crystal structure (*Raghunathan et al., 2000*), biochemical studies (*Casas-Finet et al., 1987*; *Ferrari et al., 1997*; *Raghunathan et al., 2000*), and recent all-atom MDs simulations (*Maffeo, 2015*).

In the latter, a density map of DNA on *Eco*SSB was generated from MD simulations of the protein with free nucleotides in solution. The density map was extracted from the atomic trajectory by replacing each C1′ atom on the nucleotide with a Gaussian distribution with standard deviation equal to the van der Waals radius of the atom. This process was repeated at every frame of the simulation trajectory and the result temporally averaged. The resulting density map was then spatially averaged with maps produced by rotation about each symmetry axis of the homotetramer (Maffeo, personal communication). The regions of highest DNA density were found to be located near the Trp-54, His-55, and Arg-56 residues, consistent with their known role in maintaining protein-DNA stability (*Casas-Finet et al., 1987*; *Ferrari et al., 1997*; *Raghunathan et al., 2000*) (*Figure 3—figure supplement 1*, green molecular surfaces).

Nucleotides in the wrapped ssDNA interacting with these 'hotspots' were determined based on the distance between their phosphate groups and the amino acid residues 54–56. Utilizing the SSB crystal structure, 6–7 nt per hotspot were found within a 5–7 Å distance. The set of distances, $x_{SSB}$, and number of nucleotides, $N_w$, between groups of nucleotides associated with each hotspot were then calculated and a smooth contour spanning the range of that set determined (*Figure 3—figure supplement 1* right panel, black numbered contours). The intersection points between the curves generated by *Equation 5* and the contours from the above hotspot analysis provided the tightest set of limits on $N_w$ for each data point $\Delta x(F)$ (*Figure 3—figure supplement 1* right panel, shaded colored areas). We selected the center of the range as the best estimate for $N_w$ (black dots). These served as a basis for determining the possible wrapping conformations of the complex (*Figure 3C* colored points).

## RecA-SSB competition model

The extension of ssDNA is known to increase by 50% compared to B-form dsDNA upon binding by RecA (*Hegner et al., 1999*; *Galletto et al., 2006*). Thus, the extension of the construct fully polymerized with RecA, $x_{RecA}$, is given by:

$$x_{RecA}(F) = \xi_{ds}(F) \cdot N_{ds} + 1.5\xi_{ds}(F) \cdot N_{ss}, \tag{6}$$

where $N_{ds} = 3260$ bp is the total length of the dsDNA handles and $N_{ss} = 70$ nt is that of the ssDNA loading site. Subtracting *Equation 6* from the extension of the bare DNA molecule, $x_{bare}$, given by *Equation 1*, gives the extension change:

$$\Delta x(F) = 1.5\xi_{ds}(F) \cdot N_{ss} - \xi_{ss}(F) \cdot N_{ss}, $$

which is ~10 nm at $F = 5$ pN, closely matching observations (*Figure 5—figure supplement 1*).

In measurements of RecA displacing a bound SSB (*Figure 5*), the extension change includes contributions from SSB alone, RecA with SSB, and RecA alone on ssDNA. The first and last of these are given by *Equations 2, 6*, respectively. A molecule loaded with $N_w$ nucleotides wrapped by an SSB, and the remaining $N_{ss} - N_w$ nucleotides loaded with RecA, on the other hand, has an extension:

$$x_{SSB+RecA}(F) = \xi_{ds}(F) \cdot N_{ds} + 1.5\xi_{ds}(F) \cdot (N_{ss} - N_w) + x_{SSB}^{eff}(N_w, F). \tag{7}$$

In *Figure 5D*, five distinct states are observed. These are well modeled by the following: (i) one SSB in the (SSB)$_{56}$ binding mode with no RecA bound (*Equation 2* with $N_w = 56$ nt), (ii) one SSB in the (SSB)$_{35}$ binding mode with no RecA bound (*Equation 2* with $N_w = 35$ nt), (iii) one SSB in the (SSB)$_{35}$ binding mode with all remaining unwrapped nucleotides fully loaded with RecA (*Equation 7* with $N_w = 35$ nt), (iv) one SSB in the (SSB)$_{17}$ binding mode with all remaining unwrapped nucleotides fully loaded with RecA (*Equation 7* with $N_w = 17$ nt), (v) no SSB bound, RecA fully polymerized on the ssDNA (*Equation 6*).

## Energy landscape

### Determination of wrapping intermediate energies

The energy landscape of the SSB-ssDNA nucleoprotein complex was estimated from FECs and from data of wrapping conformation vs force. First, the total free energy of wrapping, $G_{wrap}$, was estimated from the area between FECs of the protein-bound and bare DNA molecules, $x_{wrap}(F)$ and $x_{bare}(F)$ (see *Equations 1, 2* and *Figure 1*), integrated to the average SSB dissociation force. The free energy of the protein-bound DNA molecule to a force $F$ is given by:

$$G_{SSB-bound}(F) = G_{wrap} + \int_0^F x_{wrap}(F')\,dF',$$

whereas that of the bare, protein-free DNA is $G_{bare}(F) = \int_0^F x_{bare}(F')\,dF'$. Both integrals represent the free energy of stretching to force $F$. At the dissociation force $F_{1/2}$, the probabilities that an SSB is wrapped or unwrapped are equal, that is, the two free energies are equal. It follows that:

$$G_{wrap} = \int_0^{F_{1/2}} \left(x_{bare}(F') - x_{wrap}(F')\right)dF',$$

which is the negative area between the FECs in *Figure 1*.

The remaining features of the energy landscape were determined from the wrapping conformation probabilities vs force. The presence of four wrapping conformations, (SSB)$_{65}$, (SSB)$_{56}$, (SSB)$_{35}$, (SSB)$_{17}$, and an unwrapped state implies that the energy landscape is dominated by five potential wells. Applying force to the complex tilts the energy landscape (*Bustamante et al., 2004*), and changes the free-energy difference between these states. The probability the complex adopts a particular wrapping state $i$ at force $F$ is given by the Boltzmann distribution, that is,

$$p_i(F) \propto e^{-\left(G_i + G_{stretch}(F)\right)/k_B T}, \tag{8}$$

where $G_i$ is the free energy of state $i$ and $G_{stretch}(F) = \int_0^F x_i(F')\,dF'$ is the free energy of stretching the SSB-ssDNA complex in state $i$ to force $F$. The free energy difference between two states $i$ and $j$ can, therefore, be expressed as

$$\frac{p_i(F)}{p_i(F)} = e^{-\left(\Delta G_{ij} + \Delta G_{stretch}(F)\right)/k_B T}, \tag{9}$$

where $\Delta G_{ij} = G_i - G_j$ and $\Delta G_{stretch}(F) = \int_0^F (x_i(F') - x_j(F'))\,dF'$.

As described in the text, each peak in the histograms of extension change vs force in *Figure 2* was assigned a particular wrapping state $i$, as detailed in *Figure 3*. We determined the probability $p_i(F)$ from the ratio of the area under the peak to the total area in the histogram at force $F$, (*Figure 6—figure supplement 1*). From *Equation 9*, we determined the free energy difference between pairs of states, evaluating $\Delta G_{stretch}(F)$ from the area between curves of extension vs force for the two wrapping states $i$ and $j$ according to *Equation 2*. Since some of the same states were populated at different forces, we obtained several estimates of the same free energy differences. All yielded consistent values, which were averaged together and used to calculate a standard error. Setting the free energy of the unwrapped state $G_0 = 0$, the free energy associated with each state was calculated to be $G_{17} = -6.80 \pm 0.82\ k_B T$, $G_{35} = -15.38 \pm 0.57\ k_B T$, $G_{56} = -20.39 \pm 0.83\ k_B T$, and $G_{65} = -21.11 \pm 0.83\ k_B T$. The corresponding energy landscape is presented in *Figure 6*.

### Determination of barrier heights

The barrier heights for the energy landscape of the SSB-ssDNA nucleoprotein complex were estimated from lifetime measurements of the different wrapping conformations vs force as shown in *Figure 2*. The four identified wrapping conformations, (SSB)$_{65}$, (SSB)$_{56}$, (SSB)$_{35}$, (SSB)$_{17}$, and the unwrapped state undergo force-induced transitions between each other according to the following linear kinetic pathway:

$$0 \rightleftharpoons 17 \rightleftharpoons 35 \rightleftharpoons 56 \rightleftharpoons 65, \tag{10}$$

ordered from smallest to largest extension change relative to unwrapped. The rate constants for transitions between states $i$ and $j$ at a force $F$ have the form (*Dudko et al., 2008*):

$$k_{i \to j}(F) = k_0 \exp\left(-\left(\Delta G^{\ddagger} + \int_0^F \Delta x^{\ddagger}(F')\,dF'\right)/k_B T\right),$$

where $k_0$ is the attempt rate over the barrier, $\Delta G^{\ddagger}$ is the barrier height at zero force, and $\Delta x^{\ddagger}$ is the distance between state $i$ and the transition state between $i$ and $j$. The integral in the exponential

accounts for the effect of force on the barrier (*Dudko et al., 2008*). For $\Delta x^{\ddagger} > 0$, corresponding to a wrapping transition, the barrier increases with force and the rate decreases (conversely, for $\Delta x^{\ddagger} < 0$, corresponding to unwrapping, the barrier decreases and the rate increases). For example, the rate of wrapping from (SSB)$_{35}$ to (SSB)$_{56}$ is given by

$$k_{35\rightarrow56}(F) = k_0 \exp\left(\frac{\left(G^{\ddagger}_{35/56} - G_{35}\right) + \int_0^F \left(x^{\ddagger}_{35/56}(F') - x_{35}(F')\right)dF'}{k_BT}\right), \quad (11)$$

where $G_{35}$ and $x_{35}$ are the free energy and extension of the (SSB)$_{35}$ state and $G^{\ddagger}_{35/56}$ and $x^{\ddagger}_{35/56}$ are the free energy and extension of the transition state between the two wrapping states. The corresponding rate of unwrapping from (SSB)$_{56}$ to (SSB)$_{35}$ is

$$k_{56\rightarrow35}(F) = k_0 \exp\left(\frac{\left(G^{\ddagger}_{35/56} - G_{56}\right) + \int_0^F \left(x_{56}(F') - x^{\ddagger}_{35/56}(F')\right)dF'}{k_BT}\right). \quad (12)$$

Note that the equilibrium constant between the two states is

$$k^{eq}_{35\rightarrow56}(F) = \frac{k_{35\rightarrow56}(F)}{k_{56\rightarrow35}(F)} = e^{-\left(\left(G_{56}-G_{35}\right)+\int_0^F\left(x_{56}(F')-x_{35}(F')\right)dF'\right)\big/k_BT},$$

which matches *Equation 9*, as expected.

According to the pathway (10), the lifetime of the $i$-th state is given by the rates out of that state:

$$\tau_i = \frac{1}{k_{i\rightarrow i+1} + k_{i\rightarrow i-1}}.$$

In addition, the probabilities that the complex undergoes a transition from state $i$ to $i \pm 1$ are given by:

$$p_{i\rightarrow i\pm1} = \frac{k_{i\rightarrow i\pm1}}{k_{i\rightarrow i+1} + k_{i\rightarrow i-1}}.$$

Both quantities were measured directly from the constant force experiments (*Figure 2*), and the individual wrapping and unwrapping rate constants were determined from the relation $k_{i\rightarrow i\pm1} = p_{i\rightarrow i\pm1}/\tau_i$ (*Figure 6—figure supplement 2*). To determine the barrier heights, we fitted these rates to expressions of the form *Equations 11, 12*. We used a value of $k_0 \sim 10^7$ s$^{-1}$ for the attempt rate, consistent with estimates based on Kramers' kinetic theory (*Kramers, 1940*) and the range of values used in nucleosome unwrapping experiments (*Pope et al., 2005*) and protein and nucleic acid unfolding experiments (*Yang and Gruebele, 2003*; *Woodside et al., 2006*). For simplicity, we assumed the transition state extensions $x_i^{\ddagger}$ were force-independent. In addition, we used the values for the wrapping intermediate free energies $G_i$ and extensions $x_i$ obtained from analysis of the wrapping probabilities vs force, as described in the previous section.

Thus, the data in *Figure 6—figure supplement 2* were fitted globally using six parameters: $G^{\ddagger}_{35/56} = -2.9$ $k_BT$, $G^{\ddagger}_{17/35} = 6.9$ $k_BT$, $G^{\ddagger}_{0/17} = 15$ $k_BT$, measured relative to the unwrapped state energy $G_0 = 0$; and $x^{\ddagger}_{35/56} = 11.7$ nm, $x^{\ddagger}_{17/35} = 6.4$ nm, $x^{\ddagger}_{0/17} = 1.5$ nm, measured relative to the unwrapped state extension $x_0 = 0$. We estimate the error in the barrier heights to be ~3 $k_BT$, due to the uncertainty in the attempt rate $k_0$. The spatial and temporal resolution of our measurement at forces ≤1 pN did not allow an accurate determination of the transition rates between (SSB)$_{65}$ and (SSB)$_{56}$ binding modes. Presumably, the transitions are too rapid to be detected. We estimated that the barrier between those two states must be <15 $k_BT$, based on the argument that intermediates lasting >0.3 s would be detected. The corresponding energy landscape is presented in *Figure 6*. The positions of the barriers were estimated to be roughly halfway between states based on the fact that the wrapping and unwrapping transitions between those states were equally force-dependent (*Figure 6—figure supplement 2*).

## Acknowledgements

We are grateful to Christopher Maffeo in the Aksimentiev Laboratory for helpful discussions. We thank current and former members of the Chemla and Lohman Laboratories for providing help with experiments.

## Additional information

### Funding

| Funder | Grant reference | Author |
|---|---|---|
| National Science Foundation | MCB 09-52442 | Yann R Chemla |
| Burroughs Wellcome Fund | | Yann R Chemla |
| National Institutes of Health | R21 RR025341 | Yann R Chemla |
| National Institutes of Health | R01 GM030498 | Timothy M Lohman |

The funders had no role in study design, data collection and interpretation, or the decision to submit the work for publication.

### Author contributions

SS, Conception and design, Acquisition of data, Analysis and interpretation of data, Drafting or revising the article; RK, Conception and design, Acquisition of data, Drafting or revising the article; AGK, Drafting or revising the article, Contributed unpublished essential data or reagents; TML, Conception and design, Drafting or revising the article, Contributed unpublished essential data or reagents; YRC, Conception and design, Analysis and interpretation of data, Drafting or revising the article

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
