## [Decision Letter]

Thank you for submitting your work entitled “Structural dynamics of *E. coli* single-stranded DNA binding protein reveal wrapping and unwrapping pathways” for peer review at *eLife*. Your submission has been favorably evaluated by John Kuriyan (Senior Editor) and three reviewers.

The reviewers have discussed their reviews with one another and the editor has drafted this decision to help you prepare a revised submission.

This is an interesting manuscript that reports the multiple conformations of single-stranded DNA (ssDNA) wrapped on *E. Coli* ssDNA binding protein (SSB) and their interconversion dynamics. Interactions between ssDNA and SSB have been widely investigated using different experimental approaches, including single-molecule methods. In this work, Suksombat et al. applied single-molecule force and fluorescence spectroscopy with improved resolution to elucidate the conformations of ssDNA on SSB. An earlier study (Zhou et al., Cell 2011), using a similar approach, demonstrated that the interactions between SSB and ssDNA are modulated by tension applied to the nucleic acid, and that SSB tetramers likely diffuse along ssDNA by a reptation mechanism. Consistent with published findings, the authors observe deviation from a fully wrapped state (65 nt) at forces above 1 pN, indicating partial ssDNA dissociation. In constant tension experiments, the authors are able to directly observe transitions among multiple discrete states differing in extension, indicating intermediate wrapping states. Simultaneous fluorescence detection of single labeled SSB molecules indicates that transitions between these states are not due to binding of additional SSB molecules to the same DNA molecule. Comparison with structural models suggests that the distinct wrapping states are consistent with previously described states involving 17, 35, 56 and 65 nt. The experimental results presented here do not allow an unambiguous assignment to structural states. The authors' interpretation is, however, supported by experimental data obtained with a SSB mutant that has a defect in a DNA-binding “hot spot”.

smFRET experiments with labels on SSB and on the DNA under tension enabled the author to visualize wrapping/unwrapping transitions and sliding of DNA-bound SSB (Figure 4). Their results are consistent with the previously described reptation mechanism of SSB diffusion along ssDNA. Holding SSB-loaded ssDNA at 5pN in the presence of RecA revealed a step-wise displacement of SSB by RecA polymerization. Interestingly, the states observed during the process agree well with the 17, 35 and 56 nt wrapping states, lending further support to the notion that these states are actual intermediates of the (un)wrapping process. Finally, they determine an energy landscape for the ssDNA-SSB complex that shows that the unwrapping energy increases with the unraveling of the ssDNA from the protein.

The work showcases the technical merit of combined high-resolution optical trapping and fluorescence detection, and has the potential to lead to widespread applications of this technique in other complex systems. Although the work is, in principle, suitable for *eLife*, the reviewers have raised several substantial concerns that need to be addressed in a revised manuscript.

Important concerns to address:

1) One of the authors’ central findings is that ssDNA unwraps from SSB in discrete steps, not gradually (as proposed earlier). This finding implies barriers of substantial heights between the intermediates. Additional analysis (e.g. with step-finding or hidden Markov model type approaches) would reveal the force-dependent rates for transitions between individual states and, thus, the heights of the barriers separating individual states. Such insight would add value to the energy landscape presented in Figure 6. From the data presented in this study (e.g. Figure 2), it appears that such an analysis should be feasible. The reviewers feel that the calculation of these rates is important, because they may show an optimal pathway for SSB to bind and dissociate.

2) Individual SSB binding and unbinding events were resolved, as are shown in Figure 2. However, such discrete unbinding events are not seen in the force-extension curves exhibited in Figure 1 expected as rip signals. The authors are recommended to show representative traces with such rips. If these rips cannot be detected during the stretching process, please explain why.

3) In Figure 1, does the extension difference converge to zero at a zero force? If not, how was the area calculated to derive the association energy between ssDNA and SSB?

4) Figure 3: The SSB-ssDNA structure appears to be symmetric. Why does ssDNA dissociate from SSD following an asymmetric pathway as shown by the models in Figure 3?

5) In the third paragraph of the subsection “Intermediates correlate with different SSB binding modes,” the authors start to explain how they are mapping the different binding modes on to the structure of SSB. This approach is mostly displayed and explained with Figure 3—figure supplement 1. This explanation is not clear. It was not possible to judge the *x*_*SSB*_ vs *N*_*w*_ contour plots in this supplemental figure, as well as the mapping of the “force” lines onto this contour map. Given the prominent role of this figure and method, it is paramount that more effort is put into explaining it.

6) Figure 3 is not explained in the main text. It is quite important to do so because it is the raw data that leads to Figure 3. 3C, in turn, is the plot that forms the basis of the mapping of the different modes on the proteins. Figure 3 shows large error bars, in particular at low forces. The error bars of the data taken at the lowest force intercept all model lines. The other low force points do so as well to a certain extent. Oddly enough the error bars in Figure 3 have become very small for these forces. How is that possible? Moreover, from these low force points the authors conclude that they represent the fully wrapped SSB complex. That seems not really a conclusion that can be drawn on the basis of the data in Figure 3. Without that conclusion the interpretation of the other states is considerably less solid.

7) In the third paragraph of the Discussion the authors discuss that in the presence of SSB, RecA removes the SSB in a stepwise fashion as shown in Figure 5. However, the data without SSB shows also a stepwise binding of the RecA (see Figure 5—figure supplement 1). So it seems that it nearly impossible to assign steps in the RecA binding to actual removal events of the SSB. The authors should address this issue.

Other points that the authors should consider:

1) The control experiment with the SSB mutant is quite nice. However, the manuscript would benefit from a more thorough analysis of these data as well. Given the fact that they can measure the different binding states in the mutant, it should also be feasible to do an analysis that is similar to the one done in Figure 3. Such analysis should demonstrate that ssDNA maps in a similar way onto the mutant SSB.

2) To derive the conformations of different binding states, it is crucial to model the intrinsic size of the SSB-DNA complex based on the crystal structure, as is illustrated in Figure 3. The same method has previously been applied to derive protein folding intermediates (please see Figure S10 in Gao, Y. et al., Science 337, 1340-1343, 2012), which should be cited.

3) In the second paragraph of the subsection “SSB in Intermediate Wrapping States Can Diffuse on ssDNA,” the authors proposed that SSB diffuses through a reptation mechanism. Because there is no detectable extension change during the sliding process, the ssDNA may slide via small bulges. Thus, it is expected that the FRET signal accompanying the sliding process should change continuously, but not abruptly as is shown in Figure 4. Please clarify this. Can the diffusion constant be estimated from the signals? In addition, the argument to rule out the rolling mechanism of diffusion is not understood. Please revise the relevant sentences to clarify the argument.

4) Wrapping/unwrapping depends on the ionic conditions of the buffer. The experiments presented here were carried out in 100 mM Tris-HCl (pH 7.6), 10 mM NaCl, 0.1 mM EDTA. The authors claim (in the subsection “Optical tweezers experiment”) that this buffer favors (SSB)_65_ mode, but do not show data. This data should be presented in the supplement, if possible.

5) It is not always clear what the error bars represent. It may help the reader to add this information to the respective figure or figure legend.

6) In Figure 2, some of the extensions in panel B do not seem to match the extensions in panel C, particularly for the 7 pN and 9 pN data (i.e. the position of the dotted lines in B does not match the position of the peaks in C). Is there a reason for this discrepancy?

7) Readers may find it a bit confusing that part of the experimental data (the RecA experiments) is described in the Discussion section.

8) The schematic in the inset of Figure 3 is misleading/confusing, because it seems to indicate that SSB introduces an “offset” of the dsDNA portions along the vertical axis. The representation in the top panel of Figure 3 seems better.

9.. The authors do not provide a reference for the correction that they apply to model the effect of tension on the effective SSB length based on the molecule's rotational degrees of freedom. This should be better explained in the Methods section.

10). Would one expect to see discrete transitions in the traces presented in Figure 1, based on the observation of discrete transitions at constant force (Figure 2)? Perhaps the authors can comment on this point.

---

## [Author Response]

*1) One of the authors’ central findings is that ssDNA unwraps from SSB in discrete steps, not gradually (as proposed earlier). This finding implies barriers of substantial heights between the intermediates. Additional analysis (e.g. with step-finding or hidden Markov model type approaches) would reveal the force-dependent rates for transitions between individual states and, thus, the heights of the barriers separating individual states. Such insight would add value to the energy landscape presented in*
Figure 6*. From the data presented in this study (e.g.*
Figure 2*), it appears that such an analysis should be feasible. The reviewers feel that the calculation of these rates is important, because they may show an optimal pathway for SSB to bind and dissociate*.

This is a good suggestion. In the revised manuscript we describe how measurements of the dwell times and transition probabilities between different wrapping states (Figure 2) were used to determine the force-dependent rate constants for transitions between the states (plotted in a new Figure 6—figure supplement 2). From these we were able to estimate barrier heights, making reasonable assumptions for the “attempt rate” over the barrier and for the force-dependence of the transition rate. These considerations are discussed in the revised Materials and methods section.

In terms of determining an optimal pathway for SSB binding, we note that nearly every transition in our data set (373 out of 380) occurs between adjacent states in the energy landscape (Figure 6). For example, we only observe transitions between (SSB)_56_ and (SSB)_35_, and never directly between (SSB)_56_ and (SSB)_17_. As discussed in the revised text this indicates a single, linear kinetic pathway for wrapping, consistent with the energy landscape proposed.

*2) Individual SSB binding and unbinding events were resolved, as are shown in*
Figure 2*. However, such discrete unbinding events are not seen in the force-extension curves exhibited in*
Figure 1
*expected as rip signals. The authors are recommended to show representative traces with such rips. If these rips cannot be detected during the stretching process, please explain why*.

This is an important point. The rips are difficult to detect in FECs because they are small (typically <5 nm, as seen in Figure 2) and because the pulling rate is fast (∼40 nm/s, as seen in Figure 1—figure supplement 3), which smooths over transitions. As explained in the original text we believe this is also why the wrapping intermediates were not detected in earlier pulling measurements by Zhou et al. (Cell, 2011). Thus, the best way to observe these transitions experimentally is at a constant force and averaging the data to low bandwidth, as in Figure 2. Nevertheless, rips are occasionally observed in the FECs. For example, the two red traces in Figure 1 show ∼5-nm transitions at ∼5 pN, which match well with the transition between (SSB)_56_ and (SSB)_35_ modes observed in constant force experiment at 5 pN (see Figures 2 and 3: blue and green data points at 5 pN). We note that such rips are easier seen in the extension change Δ*x* vs. *F* plot rather than the raw FEC due to their small size. Some text has been added to clarify this point.

*3) In*
Figure 1*, does the extension difference converge to zero at a zero force? If not, how was the area calculated to derive the association energy between ssDNA and SSB?*

The extension difference does extrapolate to zero. This is seen by observing that the FECs for bare and SSB-bound DNA converge at low forces (Figure 1 and Figure 1—figure supplement 1). Moreover, the model to which the data are fitted to goes through the origin. This is clarified in the revised Figure 1—figure supplement 1 and caption. The wrapping energy was calculated from the area between two FECs.

*4)*
Figure 3*: The SSB-ssDNA structure appears to be symmetric. Why does ssDNA dissociate from SSD following an asymmetric pathway as shown by the models in*
Figure 3*?*

This is an important point. The SSB-ssDNA structure is actually *not* symmetric about the entry and exit points of ssDNA on the protein. This can be seen by following the DNA strands from the 5’ and 3’ ends in Figure 1. The contacts made by ssDNA with the protein are different depending which end one starts from. While the 5’ end first interacts with the proximal subunit (shown in yellow in Figure 1), the 3’ end actually makes contacts with corresponding residues in the distal subunit (red, Figure 1). We have added clarifying text in the Figure 1 caption. This rationalizes why ssDNA unravels asymmetrically in our schematics (Figure 3 and Figure 3—figure supplement 2).

*5) In the third paragraph of the subsection “Intermediates correlate with different SSB binding modes,” the authors start to explain how they are mapping the different binding modes on to the structure of SSB. This approach is mostly displayed and explained with*
Figure 3—figure supplement 1*. This explanation is not clear. It was not possible to judge the* x_SSB_
*vs* N_w_
*contour plots in this supplemental figure, as well as the mapping of the “force” lines onto this contour map. Given the prominent role of this figure and method, it is paramount that more effort is put into explaining it*.

We thank the reviewers for pointing this out. We have added extensive discussion in the main text and Materials and methods and revised Figure 3—figure supplement 1 to clarify our model. We have also switched panels 3A and 3B in the revision for clarity. As explained in this new text, each colored “force line” in Figure 3—figure supplement 1 represents all the possible values *N*_*w*_ and *x*_*SSB*_ can take for a given individual data point Δ*x* at force *F* in (revised) Figure 3. This stems from [Disp-formula equ3] and [Disp-formula equ4], which show that Δ*x(F)* is a function of both *N*_*w*_ and *x*_*SSB*_. In the revised Figure 3—figure supplement 1, we have also given each “force line” a thickness representing the error associated with the data point Δ*x(F)* that generated it (see point #6 below).

The three panels in Figure 3—figure supplement 1 use the structural data to confine the range of allowable *x*_*SSB*_ and restrict the range of *N*_*w*_ possible for each data point Δ*x* at force *F*. This is done three times in Figure 3—figure supplement 1, where we place the least (left panel) to most restrictive (right panel) limits. The least restrictive limit is simply that *x*_*SSB*_ must be <6.5 nm, the largest dimension of the SSB. At the next level (middle panel), we use the structure of ssDNA around SSB to determine the maximum and minimum *x*_*SSB*_ possible for a given *N*_*w*_. The grey shaded area in this panel thus represents the space of possible pairs of *N*_*w*_ and *x*_*SSB*_ given the structure, and the intersection between it and a force line provides the range of *N*_*w*_ consistent with the corresponding data point Δ*x* at force *F*. At the last level (right panel), we use the ‘hotspot’ analysis to restrict the space of possible pairs of *N*_*w*_ and *x*_*SSB*_ even further (black contours), giving us the tightest limits and best estimates of *N*_*w*_ for each data point Δ*x* at force *F*.

The procedure allows us to map every data point of Δ*x* vs. *F* in Figure 3 (revised) to a corresponding range of *N*_*w*_ vs. *F* in Figure 3. It is important to point out that every level of this analysis leads to the same four “bands” in Figure 3 (colored purple, blue, green, red; compare dotted lines, dashed lines, and shaded areas for the three levels of analysis), and thus basic features of our model are not so sensitive on the limits placed on *x*_*SSB*_.

*6)*
Figure 3
*is not explained in the main text. It is quite important to do so because it is the raw data that leads to*
Figure 3*. 3C, in turn, is the plot that forms the basis of the mapping of the different modes on the proteins.*
Figure 3
*shows large error bars, in particular at low forces. The error bars of the data taken at the lowest force intercept all model lines. The other low force points do so as well to a certain extent. Oddly enough the error bars in*
Figure 3
*have become very small for these forces. How is that possible? Moreover, from these low force points the authors conclude that they represent the fully wrapped SSB complex. That seems not really a conclusion that can be drawn on the basis of the data in*
Figure 3*. Without that conclusion the interpretation of the other states is considerably less solid*.

We thank the reviewers for noticing these errors. We have added text to describe this figure panel (3B in the original manuscript, now 3A in the revision). Briefly, this panel displays the values of the peaks of the Δ*x* distributions in Figure 2 vs. force. These represent the mean extension in each of the different wrapping states. In our original submission, the error bars were the standard deviations, obtained from the widths of the peaks in Figure 2. However, we now realize that the standard error of the mean (or how accurately we determine the mean extension) would be more meaningful, and have modified the figure and caption in our revision accordingly.

Note that the standard error of the mean is much smaller than the standard deviation. As a result, the low-force data (purple) is clearly more consistent with the 65 wrapping model (purple line) than any other. This result is consistent with Zhou et al. (Cell, 2011), who reported that the protein wraps 65 nt at forces ≤ 1 pN.

In addition, we have carried out a more careful analysis of the errors in this analysis. In the revised Figure 3—figure supplement 1, we now account for the standard error in Δ*x* in the “force” lines (represented as the width of the line). This changes the error bars in Figure 3 slightly, particularly at F = 0.5-1 pN, which the reviewers correctly noted were too small in the original submission.

*7) In the third paragraph of the Discussion the authors discuss that in the presence of SSB, RecA removes the SSB in a stepwise fashion as shown in*
Figure 5*. However, the data without SSB shows also a stepwise binding of the RecA (see*
Figure 5—figure supplement 1*). So it seems that it nearly impossible to assign steps in the RecA binding to actual removal events of the SSB. The authors should address this issue*.

This is an important point. While we observe pauses in the RecA-only binding traces, these do not appear at the same locations over multiple traces. As a result, the histograms in Figure 5—figure supplement 1 (right panel), which sum over ALL measurements, display no features between the initial (0 nm) and final (9 nm) pauses. In contrast, the majority of RecA+SSB traces display pauses at the same set of positions, and these thus emerge clearly in the histograms in Figure 5. To clarify this point, we have added text to the captions and additional example traces to Figure 5 and Figure 5—figure supplement 1.

*Other points that the authors should consider*:

*1) The control experiment with the SSB mutant is quite nice. However, the manuscript would benefit from a more thorough analysis of these data as well. Given the fact that they can measure the different binding states in the mutant, it should also be feasible to do an analysis that is similar to the one done in*
Figure 3*. Such analysis should demonstrate that ssDNA maps in a similar way onto the mutant SSB*.

We thank the reviewers for this suggestion. We have collected additional data and added a similar analysis for the SSB mutant. In the revised Figure 3—figure supplement 3, we show a plot of wrapping state vs. force for this mutant. The same wrapping states emerge as for wt SSB (56, 35, 17 nt), with the equilibrium shifted toward less ssDNA wrapping, and 35 nt the most likely configuration (some 56 nt wrapping intermediate is observed at the lower forces, but is much less likely). The mutant SSBs also dissociate at a lower force (∼5 pN).

*2) To derive the conformations of different binding states, it is crucial to model the intrinsic size of the SSB-DNA complex based on the crystal structure, as is illustrated in*
Figure 3*. The same method has previously been applied to derive protein folding intermediates (please see Figure S10 in Gao, Y. et al., Science 337, 1340-1343, 2012), which should be cited*.

Thank you for the suggestion. The reference has been added to the text.

*3) In the second paragraph of the subsection “SSB in Intermediate Wrapping States Can Diffuse on ssDNA,” the authors proposed that SSB diffuses through a reptation mechanism. Because there is no detectable extension change during the sliding process, the ssDNA may slide via small bulges. Thus, it is expected that the FRET signal accompanying the sliding process should change continuously, but not abruptly as is shown in*
Figure 4*. Please clarify this. Can the diffusion constant be estimated from the signals? In addition, the argument to rule out the rolling mechanism of diffusion is not understood. Please revise the relevant sentences to clarify the argument*.

To answer the first point, it is important to remember that FRET efficiency is a highly non-linear function of distance: E = 1/(1+(r/r_0_)^6^). FRET changes most rapidly at distances close to the Förster radius r_0_ = 6 nm, a range of ±2 nm about r_0_. This non-linearity results in abrupt transitions in FRET as the SSB diffuses over this distance range. At 5 pN, where the measurements were carried out, r_0_ = 6 nm ≈ 17 nt, and FRET will saturate whenever the SSB_f_ is <11 nt or >22 nt from the ss-dsDNA junction. Combined with the 0.5 Hz frequency at which the data are plotted in Figure 4, we should expect the “two-state” behavior we observe as the SSB_f_ diffuses rapidly across the short distance range of FRET. We have added text to clarify this point.

This reasoning can be used to estimate the SSB diffusion constant. To a good approximation, the lifetime of the high FRET state corresponds to the time the SSB takes to diffuse a distance equal to the Förster radius from the ss-dsDNA junction. This yields D = 15-27 nt^2^/s depending on binding mode. Roy et al. (Nature 2009) reported measurements of the diffusion coefficient across a range of temperatures (and at zero force). Using their numbers, we determine a D = 75 nt^2^/s at 23°C (our measurement temperature) and zero force. Moreover, according to Zhou et al. (Cell, 2011) a force of 5 pN will reduce D by a factor of ∼4, yielding ∼20 nt^2^/s, in very good agreement with our estimates. We have added this information to the main text.

As for our argument ruling out a rolling mechanism, we have added a new schematic clarifying this point. Figure 4—figure supplement 1 illustrates what signal we would expect for such a mechanism vs. sliding/reptation.

*4) Wrapping/unwrapping depends on the ionic conditions of the buffer. The experiments presented here were carried out in 100 mM Tris-HCl (pH 7.6), 10 mM NaCl, 0.1 mM EDTA. The authors claim (in the subsection “Optical tweezers experiment”) that this buffer favors (SSB)*_*65*_
*mode, but do not show data. This data should be presented in the supplement, if possible*.

We have added the biochemical data to which the reviewers are referring. A new Figure 1—figure supplement 6 shows a binding isotherm at high (100 mM) Tris-HCl measuring fluorescence from a Cy3-Cy5 fluorescently labeled (dT)_70_ oligonucleotide with SSB. The data show that a single SSB wraps (dT)_70_ in (SSB)_65_ mode whenever [SSB] ≤ [(dT)_70_].

*5) It is not always clear what the error bars represent. It may help the reader to add this information to the respective figure or figure legend*.

We have added this information wherever appropriate. Please see the revised captions for Figure 1, Figure 3, Figure 1—figure supplement 1.

*6) In*
Figure 2*, some of the extensions in panel B do not seem to match the extensions in panel C, particularly for the 7 pN and 9 pN data (i.e. the position of the dotted lines in B does not match the position of the peaks in C). Is there a reason for this discrepancy?*

At 7 pN, the change in extension of all three states in panel B (approximately -8 nm, -10.5 nm, -13 nm) matches very well with the extension change distributions in panel C. At 9 pN, the discrepancy comes from the fact that the distribution is a collection of many data traces. The -7 nm extension state is rare, as seen from the fact that the corresponding peak in the distribution is small. The representative trace we had selected in the original manuscript displayed only two states. To avoid confusion, we selected a different trace in the revised Figure 2 that displays all three states.

*7) Readers may find it a bit confusing that part of the experimental data (the RecA experiments) is described in the Discussion section*.

While we appreciate the reviewers’ comment, we feel these data are best introduced during the discussion of binding intermediates and unwrapping pathway.

*8) The schematic in the inset of*
Figure 3
*is misleading/confusing, because it seems to indicate that SSB introduces an “offset” of the dsDNA portions along the vertical axis. The representation in the top panel of*
Figure 3
*seems better*.

We thank the reviewers for pointing this out. We have made an adjustment to the revised Figure 3.

*9) The authors do not provide a reference for the correction that they apply to model the effect of tension on the effective SSB length based on the molecule's rotational degrees of freedom. This should be better explained in the Methods section*.

A reference on the effect of tension on the SSB length has been added to the text, and a derivation included in the Materials and methods section.

*10) Would one expect to see discrete transitions in the traces presented in*
Figure 1*, based on the observation of discrete transitions at constant force (*Figure 2*)? Perhaps the authors can comment on this point*.

Please see our answer above under “Important concern #2”.